# Vesicular release probability sets the strength of individual Schaffer collateral synapses

Céline D. Dürst [1,5], J. Simon Wiegert [1,2], Christian Schulze[1], Nordine Helassa [3,4], Katalin Török [3] & Thomas G. Oertner [1] ✉

Information processing in the brain is controlled by quantal release of neurotransmitters, a tightly regulated process. From ultrastructural analysis, it is known that presynaptic boutons along single axons differ in the number of vesicles docked at the active zone. It is not clear whether the probability of these vesicles to get released ($p_{ves}$) is homogenous or also varies between individual boutons. Here, we optically measure evoked transmitter release at individual Schaffer collateral synapses at different calcium concentrations, using the genetically encoded glutamate sensor iGluSnFR. Fitting a binomial model to measured response amplitude distributions allowed us to extract the quantal parameters $N$, $p_{ves}$, and $q$. We find that Schaffer collateral boutons typically release single vesicles under low $p_{ves}$ conditions and switch to multivesicular release in high calcium saline. The potency of individual boutons is highly correlated with their vesicular release probability while the number of releasable vesicles affects synaptic output only under high $p_{ves}$ conditions.

The conceptual framework of quantal synaptic transmission was developed at the frog neuromuscular junction (NMJ), a giant synapse containing thousands of transmitter vesicles[1]. Statistical analysis of electrophysiological recordings from innervated muscle cells suggested that neurotransmitter is released in multi-molecular packets ('quanta') which were later identified as synaptic vesicles. The strength of a specific NMJ could be mathematically described[2] as the product of the number of release sites $N$, their probability of release $p_{ves}$, and the postsynaptic response to the release of a single quantum, $q$. Applying quantal analysis to the very small synapses of the mammalian brain is not straightforward. In the brain, each neuron receives thousands of synapses, each of which has a different electrotonic distance from the cell body. Viewed from the soma, there is no unitary response: a single vesicle released at a distal dendritic synapse creates a much smaller EPSP than a vesicle released at a perisomatic synapse, making classical electrophysiological quantal analysis impossible. Attempts to

electrically stimulate individual synapses gave rise to the hypothesis that individual synapses in the central nervous system can only release a single vesicle ('uniquantal release')[2]. Graded postsynaptic responses were not considered hallmarks of multivesicular release events, but attributed to a failure of single-synapse stimulation[3]. As electrophysiology does not provide spatial information about the origin of the signals, it was not possible to distinguish between the two rival interpretations at the time, with the exception of specialized circuits that are limited to a single synaptic contact[4].

Optical methods based on fluorescent calcium indicators allow resolving excitatory postsynaptic calcium transients (EPSCaTs) at individual synapses. Under conditions of high release probability, larger EPSCaTs were observed[5]. However, the amplitude of spine calcium transients depends on the local membrane potential in a highly nonlinear fashion. The low number and stochastic behavior of postsynaptic NMDA receptors[6] adds variability to EPSCaTs, making it

[1]Institute for Synaptic Physiology, Center for Molecular Neurobiology Hamburg (ZMNH), 20251 Hamburg, Germany. [2]Research Group Synaptic Wiring and Information Processing, Center for Molecular Neurobiology Hamburg (ZMNH), 20251 Hamburg, Germany. [3]Cell Biology and Genetics Research Centre, Molecular and Clinical Sciences Research Institute, St George's, University of London, London SW17 0RE, UK. [4]Department of Cardiovascular and Metabolic Medicine, Institute of Life Course and Medical Sciences, University of Liverpool, Liverpool L69 3BX, UK. [5]Present address: Department of Basic Neurosciences, Center for Neurosciences (CMU), University of Geneva, 1211 Geneva, Switzerland. ✉e-mail: thomas.oertner@zmnh.uni-hamburg.de

difficult to draw conclusions about vesicular release statistics from EPSCaT amplitude distributions. Furthermore, experimental manipulation of divalent ion concentration ($Ca^{2+}$, $Mg^{2+}$, $Sr^{2+}$) affect both the release machinery and the response of the optical calcium sensor, making the results of ion substitution experiments difficult to interpret. In dissociated neuronal culture, it is possible to monitor vesicle fusion events using pH-sensitive green fluorescent proteins (pHluorins)[7,8]. A limitation of pH-based fusion detection is the lack of information about the filling state of the released vesicles, which is one determinant of $q$[9].

Here we use the genetically encoded glutamate sensor iGluSnFR[10] to measure glutamate concentrations in the synaptic cleft[11,12]. We show that this probe is sensitive enough to detect the fusion of single vesicles at Schaffer collateral boutons in organotypic hippocampal cultures. At near-physiological calcium concentrations[13], synapses often failed to release glutamate and otherwise released single vesicles in response to single action potentials (APs). Elevating the extracellular $Ca^{2+}$ concentration caused synapses to increase their release probability and to release multiple vesicles at once. By localizing the fusion site on the surface of the presynaptic bouton with high precision, we show that multivesicular release occurs in a confined area, the active zone, which is stable over time. Based on dual patch-clamp recordings and Monte Carlo simulations of glutamate diffusion, we conclude that the dynamic range of iGluSnFR is similar to postsynaptic AMPA receptors, although the kinetics of the underlying glutamate transients in the synaptic cleft is 2–3 orders of magnitude faster than the iGluSnFR response. Thus, iGluSnFR signals are a good proxy for postsynaptic responses, but do not report the speed of glutamate diffusion out of the synaptic cleft. Comparing quantal parameters across boutons, we show that $p_{ves}$ is the main determinant of synaptic strength under low release probability conditions. Under conditions of high release probability, $p_{ves}$ and $N$ jointly determine synaptic output.

## Results

### Modulating synaptic release probability affects cleft glutamate concentration

We transfected individual CA3 pyramidal neurons in organotypic slices of rat hippocampus with iGluSnFR and the red fluorescent protein tdimer2 via single-cell electroporation. Two to four days after transfection, we transferred the cultures to the recording chamber (33 °C) of a two-photon microscope. As iGluSnFR is relatively dim in the absence of glutamate, the red cytoplasmic fluorescence was helpful to visualize soma, axon and boutons of transfected CA3 pyramidal cells. To evoke release, we patch-clamped transfected cells and triggered single APs while imaging bouton fluorescence in CA1 stratum radiatum. Boutons belonging to the patched CA3 neuron were easily identified by a rapid increase in green fluorescence, milliseconds after the somatically-triggered AP. The fast rise and decay kinetics of iGluSnFR[10,14] made it challenging to capture the peak of iGluSnFR fluorescence transients using the relatively slow raster scanning mode. To capture the peak of the iGluSnFR signal, we used spiral scans[11] to sample the entire surface of the bouton every two milliseconds (Fig. 1a, left). As we did not know the exact location of the fusion site a priori, we evaluated the spatial positions (columns in the time-space diagram) with the highest change in fluorescence[11]. If no clear stimulus-evoked change in fluorescence was detected (potential failure trial), we evaluated the same region of interest as in the last trial. To compensate for slow drift of the tissue during the experiment, we developed an automatic 3D repositioning routine, allowing stable optical recordings over hundreds of trials (Fig. 1b and Supplementary Fig. 1). Boutons with excessively noisy baseline fluorescence (>0.4 $\Delta F/F_0$) were excluded from further analysis (Supplementary Fig. 2).

To extract the amplitude of responses, we constructed a synapse-specific template (exponential decay) from several manually selected large responses. A single free parameter (amplitude) was used to match the template to each individual trial. Responses that exceeded $2\sigma$ of baseline fluctuations were classified as successes. This threshold produced consistent results with our more complex quantal analysis (Supplementary Fig. 3). Occasionally, we observed green fluorescent particles moving through the axon. Such events were detected by their elevated $F_0$ at baseline and discarded (~2–5% of trials). At most

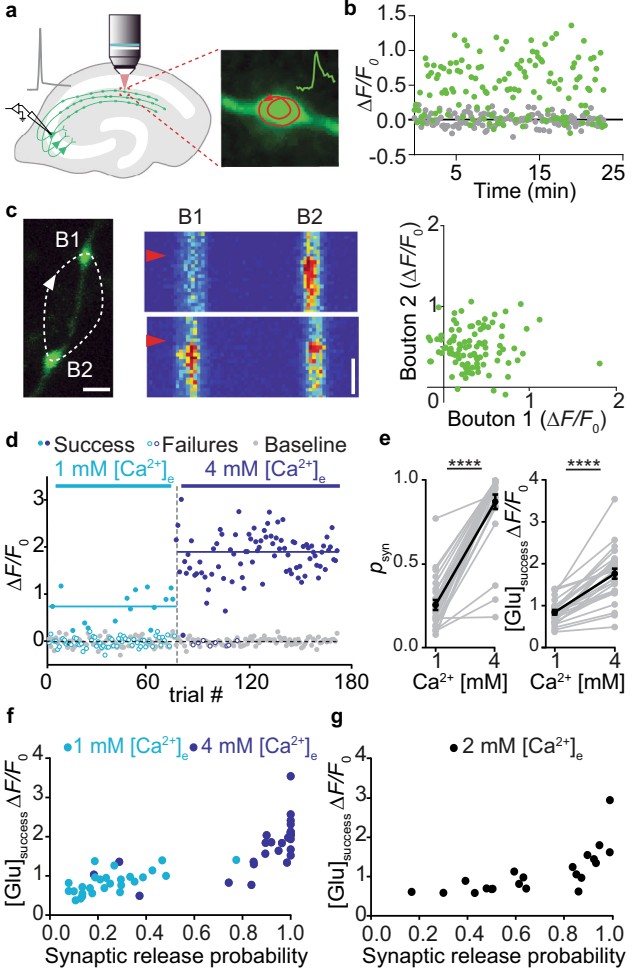

**Fig. 1 | Increased release probability increases the glutamate concentration of synaptic transmission events. a** CA3 pyramidal neuron in organotypic slice culture was patch-clamped to somatically trigger single action potentials while imaging with 2-photon laser scanning microscopy bouton fluorescence in CA1 stratum radiatum. **b** Response amplitude (green circles) monitored in artificial cerebrospinal fluid (ACSF) containing 2 mM $[Ca^{2+}]_e$ at 33 °C was constant over time. Baseline fluorescence before stimulation was analyzed to estimate imaging noise (gray circles). **c** Monitoring two neighboring boutons located on the same axon. Scale bar, 1 μm. Two trials showing independent release events. Red arrowheads indicate the stimulation onset. Scale bar, 20 ms. iGluSnFR transients between neighboring boutons were not correlated (two-sided Spearman correlation: $r = −0.04$, $p = 0.69$). **d** Glutamate transients in a single bouton, switching from ACSF containing 1 mM $[Ca^{2+}]_e$ to 4 mM $[Ca^{2+}]_e$. The light and the dark blue line represent the average amplitude of successes in 1 mM $[Ca^{2+}]_e$ and 4 mM $[Ca^{2+}]_e$, respectively. **e** Summary of all $[Ca^{2+}]_e$ switching experiments. The probability of successful glutamate release (left panel) increased from 0.26 ± 0.03 in 1 mM $[Ca^{2+}]_e$ to 0.87 ± 0.04 in 4 mM $[Ca^{2+}]_e$ (two-sided Wilcoxon-signed rank test, $p < 0.0001$, $n = 27$ boutons in 23 slices). The amplitude of fluorescence transients in trials classified as 'success' (right panel) increased from 0.84 ± 0.056 $\Delta F/F_0$ in 1 mM $[Ca^{2+}]_e$ to 1.76 ± 0.12 $\Delta F/F_0$ in 4 mM $[Ca^{2+}]_e$ (paired $t$ test, $p < 0.0001$, $n = 27$ boutons in 23 slices). Values are given as mean ± SEM. **f** The probability of release was correlated with the amplitude of success trials in a non-linear fashion. **g** In 2 mM $[Ca^{2+}]_e$, the release probability of individual synapses ranged from 0.17 to 0.99. The amplitude of success trials was similar at all boutons with $p_{syn} < 0.5$. Source data are provided as a Source Data file.

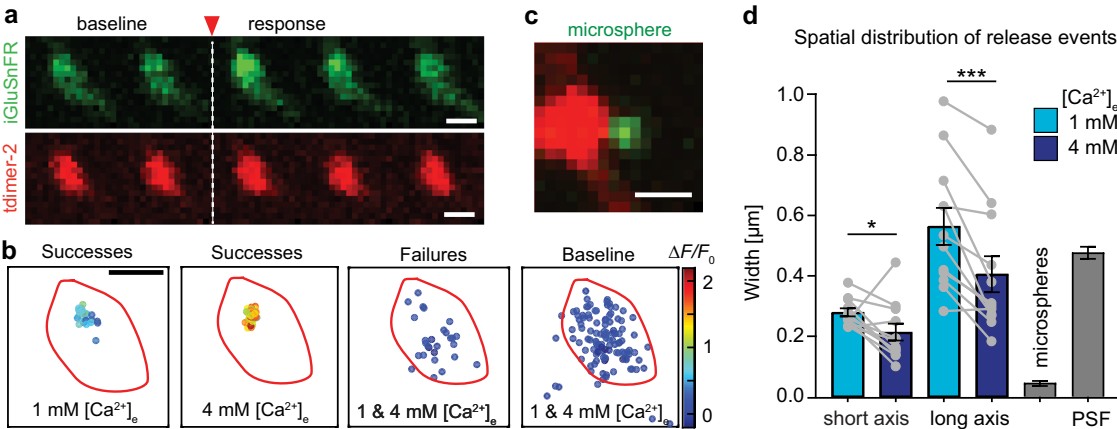

**Fig. 2 | Localizing release events at Schaffer collateral boutons. a** Rapid frame scans (62.5 Hz) show increase in iGluSnFR fluorescence (green) after stimulation (single trial, single AP triggered between frame #2 and #3). Scale bar: 1 μm. **b** The center position and amplitude (color scale) of Gaussian fits to the change in green fluorescence are plotted with respect to the bouton outline (red line). Failure trials and random changes in baseline fluorescence show no local clustering. **c** For calibration purposes, green fluorescent microspheres (0.17 μm) were imaged next to boutons. **d** To quantify the spatial distribution of release events at each bouton, an ellipse was fit to contain 95% of success locations. Both short and long axis of the ellipse were significantly smaller (short axis, two-sided Wilcoxon test, $p = 0.04$, long axis, two-sided Wilcoxon test, $p = 0.001$, $n = 12$ boutons) under high $Ca^{2+}$ conditions (short axis: $0.21 \pm 0.03$ μm; long axis: $0.41 \pm 0.06$ μm) compared to 1 mM $[Ca^{2+}]_e$ (short axis: $0.28 \pm 0.01$ μm; long axis: $0.56 \pm 0.06$ μm). For comparison, microsphere localization precision ($0.05 \pm 0.01$ μm) and xy-size of the point spread function (PSF) are also shown (gray bars, mean ± SEM). Source data are provided as a Source Data file.

boutons, the failure rate was stable over the time of the experiment (Fig. 1b and Supplementary Fig. 1). In principle, failure of glutamate release could be due to the stochastic nature of vesicle release or due to stochastic failures of AP propagation into individual branches of the extensive network of axon collaterals. In simultaneously imaged neighboring boutons, failures were not correlated, arguing for stochastic glutamate release (Fig. 1c).

Since vesicle fusion is $Ca^{2+}$-dependent, we expected a steep dependence of $p_{syn}$ on the extracellular $Ca^{2+}$ concentration, $[Ca^{2+}]_e$. Indeed, switching $[Ca^{2+}]_e$ from 1 mM to 4 mM (Fig. 1d) dramatically increased $p_{syn}$ from 0.26 to 0.87 (Fig. 1e), with some boutons reaching the ceiling of $p_{syn} = 1$. The variability of $p_{syn}$ was much larger in 1 mM $[Ca^{2+}]_e$ (CV = 0.61) compared to the same synapses in 4 mM $[Ca^{2+}]_e$ (CV = 0.26). The amplitude of iGluSnFR signals (successes) increased as well from 0.84 to 1.76 $\Delta F/F_0$ (Fig. 1e), indicating higher glutamate concentrations in the synaptic cleft under high $p_{syn}$ conditions. In low $[Ca^{2+}]_e$, success amplitudes were similar across boutons (Fig. 1f). In high $[Ca^{2+}]_e$, however, the same set of boutons had variable success amplitudes that were strongly correlated with $p_{syn}$. To further explore the non-linear relationship between $p_{syn}$ and cleft glutamate concentrations, we performed a set of experiments in 2 mM $[Ca^{2+}]_e$ (Fig. 1g). Under these conditions, $p_{syn}$ was highly variable between individual boutons. Again, low $p_{syn}$ boutons produced consistent success amplitudes (58% to 89% $\Delta F/F_0$ for $p_{syn} < 0.5$) while high $p_{syn}$ boutons produced considerably larger successes.

Next, we tested the possibility that release may not be confined to a single active zone, in particular under high $[Ca^{2+}]_e$ conditions. Using fast raster scans of individual boutons at high zoom, we generated glutamate maps that allowed us to estimate the release position on the bouton by fitting a 2D Gaussian function (Fig. 2a). The centroids of high amplitude responses ('successes') were clustered to a small region of the bouton, which was not the case for low amplitude responses ('failures') or 2D Gaussian fits to the fluctuations in baseline fluorescence (Fig. 2b). The resolution of our localization procedure was ~50 nm, estimated by imaging fluorescent microspheres next to boutons (Fig. 2c). Interestingly, the spatial distribution of glutamate release events was significantly more confined in 4 mM $[Ca^{2+}]_e$ compared to the same boutons in 1 mM $[Ca^{2+}]_e$ (Fig. 2d). As large-amplitude events in 4 mM $[Ca^{2+}]_e$ likely result from the fusion of several vesicles at once, we were actually localizing the centroids of multivesicular

events, which are expected to be less variable in space than individual fusion events. The higher photon count may have further improved localization precision. We found few examples of boutons that appeared to have multiple active zones (Supplementary Fig. 4); such multisynaptic boutons were excluded from further analysis. Taken together, our data indicate that $p_{syn}$ is very variable between boutons and highly dependent on $[Ca^{2+}]_e$. As $p_{syn}$ increased, so did the reported glutamate concentration, consistent with the capacity of Schaffer collateral boutons for multivesicular release[15–20].

## Desynchronized release events reveal quantal size
In earlier studies, competitive antagonists of AMPA receptors were used to test the possibility of multivesicular release at Schaffer collateral synapses[15,21]. To perform a classical quantal analysis, however, the size of the quantum ($q$) has to be known. To demonstrate multivesicular release directly, it is necessary to compare the amplitude of evoked responses to the amplitude of spontaneous fusion events ('minis') at the same synapse. We therefore replaced extracellular $Ca^{2+}$ with $Sr^{2+}$ to desynchronize vesicle fusion events[22] while monitoring glutamate transients at indivdual boutons (Fig. 3a). $Sr^{2+}$ is known to lead to asynchronous release due to its low affinity for synaptotagmin-1[23] and its slow clearance from the presynaptic terminal[24]. As expected, large-amplitude glutamate release events occurred with high probability in 4 mM $[Ca^{2+}]_e$. When artificial cerebrospinal fluid (ACSF) containing 4 mM $Ca^{2+}$ was slowly replaced by ACSF containing 4 mM $Sr^{2+}$, evoked glutamate transients started to disintegrate into smaller events of relatively uniform amplitude. When 4 mM $Sr^{2+}$ was fully washed-in, evoked responses completely disappeared while baseline fluorescence became very noisy. The amplitude histograms show clear separation between evoked responses (Fig. 3b, c, blue bars) and delayed events (green bars). We interpret this sequence of events during wash-in as evoked multivesicular release, delayed fusion events of individual vesicles under the influence of $[Sr^{2+}]_e$, and finally, incomplete fusion in the absence of $[Ca^{2+}]_e$. The average quantal amplitude determined with this method was $q = 82\% \pm 22\%$ $\Delta F/F_0$ ($n = 7$ boutons).

## The dynamic range of postsynaptic responses
Our optical measurements demonstrate a steep dependence of release probability on $[Ca^{2+}]_e$. Do postsynaptic AMPA receptors report increased glutamate concentrations as larger currents? To measure

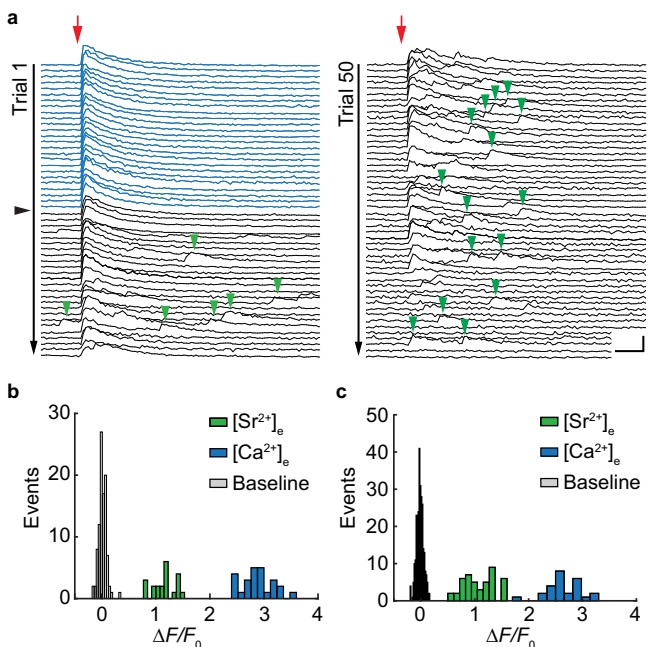

**Fig. 3 | Desynchronized glutamate release events reveal quantal size. a** iGluSnFR changes in fluorescence (single bouton) in response to single APs in ACSF containing 4 mM $[Ca^{2+}]_e$ (blue traces). Red arrow indicates stimulation onset. Synaptic release probability was 1 (no failures). The black arrowhead indicates the start of 4 mM $[Sr^{2+}]_e$ wash-in. During wash-in of 4 mM $[Sr^{2+}]_e$ discrete events appear in the wake of the stimulation (green arrowheads). Scale bars: 20 ms, 2 $\Delta F/F_0$. **b** Amplitude histogram of evoked responses measured in 4 mM $[Ca^{2+}]_e$ (blue bars) and delayed events measured in 4 mM $[Sr^{2+}]_e$ (green bars) and fluctuation of baseline fluorescence (gray bars) of the example shown in **a**. **c** Second example. Baseline fluctuations are plotted with narrow bin size to show their normal distribution. Source data are provided as a Source Data file.

the strength of unitary connections, we performed dual patch-clamp recordings from connected pairs of CA3 and CA1 pyramidal cells under NMDAR block (10 μM CPPene, Fig. 4a). For each pair of neurons, we recorded EPSCs in 1 mM and in 4 mM $[Ca^{2+}]_e$, switching the sequence (low to high/high to low $[Ca^{2+}]_e$) between experiments. The boosting of AMPA EPSCs by high $[Ca^{2+}]_e$ was similar to the boosting of iGluSnFR signals (Fig. 4c), suggesting that AMPARs were able to transmit the very high transmitter concentrations reached in 4 mM $[Ca^{2+}]_e$ (Fig. 4b). Reducing AMPAR occupancy by the competitive antagonist γ-DGG (10 mM) did not increase the difference between low and high $[Ca^{2+}]_e$ EPSCs, suggesting that AMPARs were not saturated by the glutamate released after single presynaptic APs in 4 mM $[Ca^{2+}]_e$ (Supplementary Fig. 5). We verified that blocking NMDARs did not affect the release of glutamate (Supplementary Fig. 6). Furthermore, in paired recordings, expression of iGluSnFR or a membrane-bound GFP in the postsynaptic cell did not change synaptic strength or paired-pulse ratio (Supplementary Fig. 7).

We were interested how the dramatically boosted synaptic transmission in 4 mM $[Ca^{2+}]_e$ would change short-term plasticity at this synapse. We compared paired-pulse ratios from patch-clamp recordings of connected CA3-CA1 pairs and iGluSnFR signals from individual boutons. In 1 mM $[Ca^{2+}]_e$, EPSCs showed paired-pulse facilitation (PPR = 156%), which was absent in 4 mM $[Ca^{2+}]_e$ (PPR = 86%, Fig. 4d, f). iGluSnFR responses showed weak facilitation in 1 mM $[Ca^{2+}]_e$ (PPR = 110%) and depression in 4 mM $[Ca^{2+}]_e$ (PPR = 79%, Fig. 4e, f). It is remarkable that Schaffer collateral synapses are able to maintain a fairly linear paired pulse response over a ~10-fold modulation in synaptic strength. As previously suggested, the depletion of the readily releasable pool of vesicles in high $Ca^{2+}$ conditions seems to be largely compensated by $Ca^{2+}$-dependent facilitation of release of the

remaining vesicles[25]. During prolonged high frequency trains, however, this compensatory mechanism runs out of resources, resulting in strong depression[26].

## Non-linear response of iGluSnFR to glutamate release

Fusion of a single vesicle injects a bolus of concentrated glutamate into the synaptic cleft that disperses within microseconds through diffusion. To explore how diffusing glutamate molecules interact with iGluSnFR and postsynaptic AMPARs, we implemented a Monte Carlo simulation consisting of a glutamatergic bouton contacting a dendritic spine surrounded by astrocytes. Simulated fusion of a single transmitter vesicle (containing 3000 molecules of glutamate[27]) in the center of the synaptic cleft produced a local cloud of glutamate that filled the entire cleft within 10 μs (Fig. 5a). Consequently, iGluSnFR molecules became bound (and highly fluorescent) and AMPARs opened (Fig. 5b). The model allowed us to explore how different orientations of the synapse with respect to the optical axis would affect the amplitude of iGluSnFR signals (Fig. 5c, d). The largest signal in response to fusion of a single vesicle (quantal amplitude, $q$) was generated when both spine and axon were in the focal plane, aligning the synaptic cleft with the optical axis. Model synapses where spine or axon were tilted with respect to the focal plane produced slightly smaller signals. The tilt increased the number of extrasynaptic iGluSnFR molecules within the point spread function (PSF), increasing baseline fluorescence ($F_0$) and thus decreasing the relative change in fluorescence ($\Delta F/F_0$) after simulated release of a single vesicle ($q$). In organotypic slice cultures, most spines are horizontally oriented[28], and we selected horizontal axonal sections for imaging. Nevertheless, to account for variable spine orientations, we considered $q$ as a free parameter in our quantal analysis. We used the model to test whether the orientation of the synapse would affect the saturation of iGluSnFR responses. For a specific number of simultaneously released vesicles (1–15), we performed 100 Monte Carlo simulations and plotted the peak iGluSnFR responses (Fig. 5e). Due to iGluSnFR saturation, the spacing of the Gaussian-like amplitude distributions became closer with increasing vesicle numbers. Individual peaks appear well separated as measurement noise (photon shot noise) was not part of the simulation. The shape of the saturation curve (Fig. 5f) could be approximated by a hyperbolic function

$$r = \frac{B_{max}[Glu]}{k_d + [Glu]} \tag{1}$$

where $r$ is the fraction of iGluSnFR molecules bound to glutamate, $B_{max}$ the relative change in fluorescence at saturation, and $K_d$ the apparent affinity of the iGluSnFR-expressing bouton for glutamate (i.e. the number of simultaneously released vesicles leading to 50% saturation). For a synapse at optimal orientation, we found $B_{max} = 5.9$ $\Delta F/F_0$ and $K_d = 6.5$ vesicles. At 40° spine tilt, we found $B_{max} = 7.0$ $\Delta F/F_0$ and $K_d = 10$ vesicles. iGluSnFR itself has $B_{max} = 4.4$ $\Delta F/F_0$ when calibrated with glutamate-containing solutions (which is also the case for our simulated iGluSnFR).

When measured at a synapse, however, iGluSnFR molecules outside the synaptic cleft are exposed to strongly diluted glutamate transients. The more extrasynaptic iGluSnFR molecules are within the PSF of the microscope, the lower the apparent affinity of a bouton for glutamate. We saw comparable effects when we simulated a microscope with lower resolution (larger PSF). Thus, the fluorescence increase in response to the release of a single vesicle ($q$) and the shape of the saturation curve depend not only on the biophysical properties of the indicator, but also on the spatial resolution of the microscope and on the orientation of the synaptic cleft. For optical quantal analysis of synaptic responses, it is important to consider that the apparent affinity of an iGluSnFR-expressing bouton is lower than expected from iGluSnFR molecular properties.

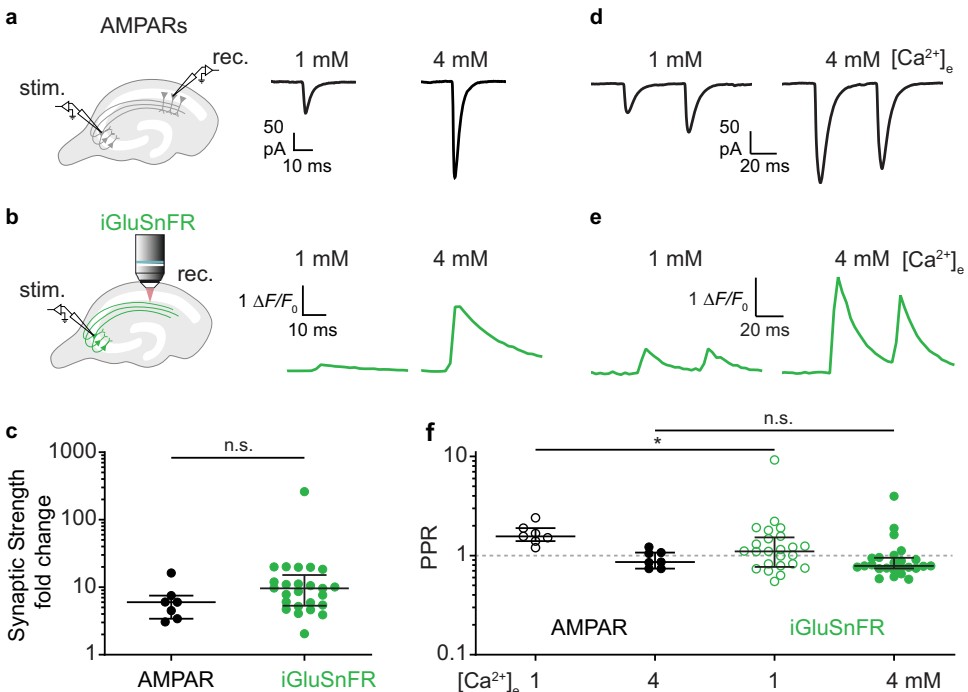

**Fig. 4 | Multivesicular release increases the gain and the signal to noise ratio of synapses. a** EPSCs were measured by dual patch-clamp recordings from connected CA3-CA1 pyramidal cell pairs. Perfusion was switched from 1 mM to 4 mM $[Ca^{2+}]_e$, leading to increased EPSC amplitudes. Traces show EPSCs (average of ~40 trials) from the CA1 pyramidal cell. **b** Traces show the evoked change in fluorescence (average of ~60 trials) from a single bouton in CA1 stratum radiatum. The switching sequence (low - high $[Ca^{2+}]_e$ / high - low $[Ca^{2+}]_e$) was reversed between experiments. **c** Increasing $[Ca^{2+}]_e$ from 1 to 4 mM increased the median amplitude of AMPAR-mediated EPSCs by a factor of 6.02, interquartile range (IQR): 3.4–7.5-fold, $n = 7$ pairs, the median iGluSnFR response by 9.63-fold, IQR: 5.3 –15.2-fold, $n = 25$ boutons. There is no significant difference between the fold change in AMPARs and iGluSnFR responses (two-sided Mann–Whitney, $p = 0.08$). **d** Representative EPSCs (average of ~40 trials) in response to the presynaptic paired-pulse stimulation (ISI

48 ms) recorded in 1 mM and 4 mM $[Ca^{2+}]_e$. **e** Representative iGluSnFR signals (average of ~60 trials) from a bouton in CA1 in response to the paired-pulse stimulation. **f** EPSCs showed paired-pulse facilitation in 1 mM $[Ca^{2+}]_e$, (median PPR: 156%, IQR: 140%–189%, $n = 7$ pairs) and paired-pulse depression in 4 mM $[Ca^{2+}]_e$ (median PPR: 86%, IQR: 74%–107%, $n = 7$ pairs). iGluSnFR responses showed paired-pulse facilitation in 1 mM $[Ca^{2+}]_e$ (median PPR: 110%, IQR: 77%–152%, $n = 25$ boutons) and paired-pulse depression in 4 mM $[Ca^{2+}]_e$ (median PPR: 79%, IQR: 74%–95%, $n = 25$ boutons). There is a significant difference between the AMPARs PPR and the iGluSnFR PPR in 1 mM $[Ca^{2+}]_e$ (two-sided Mann–Whitney test, $p = 0.03$). However, there is no significant difference between the AMPARs PPR and the iGluSnFR PPR in 4 mM $[Ca^{2+}]_e$ (two-sided Mann–Whitney test, $p = 0.37$). Value with a PPR = 0 is not plotted for display purpose but was used for statistics. Values are plotted as median with IQR. Source data are provided as a Source Data file.

## Extracting synaptic parameters by histogram analysis

The histograms of iGluSnFR responses from individual boutons often showed multiple peaks. Distinct quantal peaks have been observed in EPSC distributions from CA3 pyramidal cells in response to mossy fiber stimulation[29], but are much less clear at CA3-CA1 connections[30]. If these peaks indicate the simultaneous release of two or more vesicles in response to a single presynaptic AP, there are clear predictions about the amplitude and spacing of the peaks: The amplitude of the peaks would be expected to follow binomial statistics, as famously shown for endplate potentials[31]. Due to the saturation of iGluSnFR, however, quantal peaks should not be equidistant, but compressed according to a hyperbolic saturation function (Eq. 1).

To investigate release statistics in more detail, we performed optical quantal analysis from the iGluSnFR signals of individual Schaffer collateral boutons monitored in 1 and 4 mM $[Ca^{2+}]_e$. The fluorescence trace in every trial was fit with a kernel (exponential decay function) to extract the peak amplitude. To extract the three quantal parameters $N$, $p_{ves}$, and $q$ from the response histograms, we generated predictions (probability density functions) for all possible parameter triplets (exhaustive search) to find the combination of parameters that best fit the histogram of iGluSnFR signals. First, for every combination of $N$ and $p_{ves}$, we calculated the binomial probabilities for the different outcomes (failures, univesicular and multivesicular events, Fig. 6a). From the baseline fluorescence distribution of the synapse in question, we extracted the expected variability of failure fluorescence (width of the Gaussian). As photon shot noise increases with the square root of

the number of detected photons, we added appropriate amounts of 'noise' to the expected quantal peaks, leading to a broadening of the individual Gaussians as the signal increased. To account for gradual saturation of iGluSnFR at increasing glutamate concentrations, we spaced the expected quantal peaks not as integer multiples of $q$, but according to the saturation function that provided the best fit in our simulated experiments (Fig. 5f). For every prediction, we scaled the amplitude to match the number of observations (histogram) and calculated the mean square error. We observed that different combinations of $N$ and $p_{ves}$ generated near-identical fits, as there was no 'cost' associated with increasing $N$ in the model. We decided to select the prediction with the smallest number of vesicles that was within 2% of the minimum mean square error as the most parsimonious biophysical mechanism for the synapse in question. To further constrain the fitting procedure, the algorithm had to find values for $q$ and for $N$ that could account for the histogram of responses in 1 mM $[Ca^{2+}]_e$ and for the histogram measured in 4 mM $[Ca^{2+}]_e$. The quantal size and number of release-ready vesicles are not expected to change with $[Ca^{2+}]_e$. Only $p_{ves}$ was allowed to vary between the low and high $[Ca^{2+}]_e$ condition. The fitting results provided a convincing explanation why some boutons showed multiple peaks in 4 mM $[Ca^{2+}]_e$ while those multiple peaks were not apparent in 1 mM $[Ca^{2+}]_e$ (Fig. 6b, c, upper panels). Due to partial saturation of iGluSnFR at high glutamate concentrations, quantal peaks for three or more simultaneously released vesicles are not resolved, but compressed into a broad peak (Fig. 6b, c, lower panels). Extraction of quantal parameters was robust to the bin size of

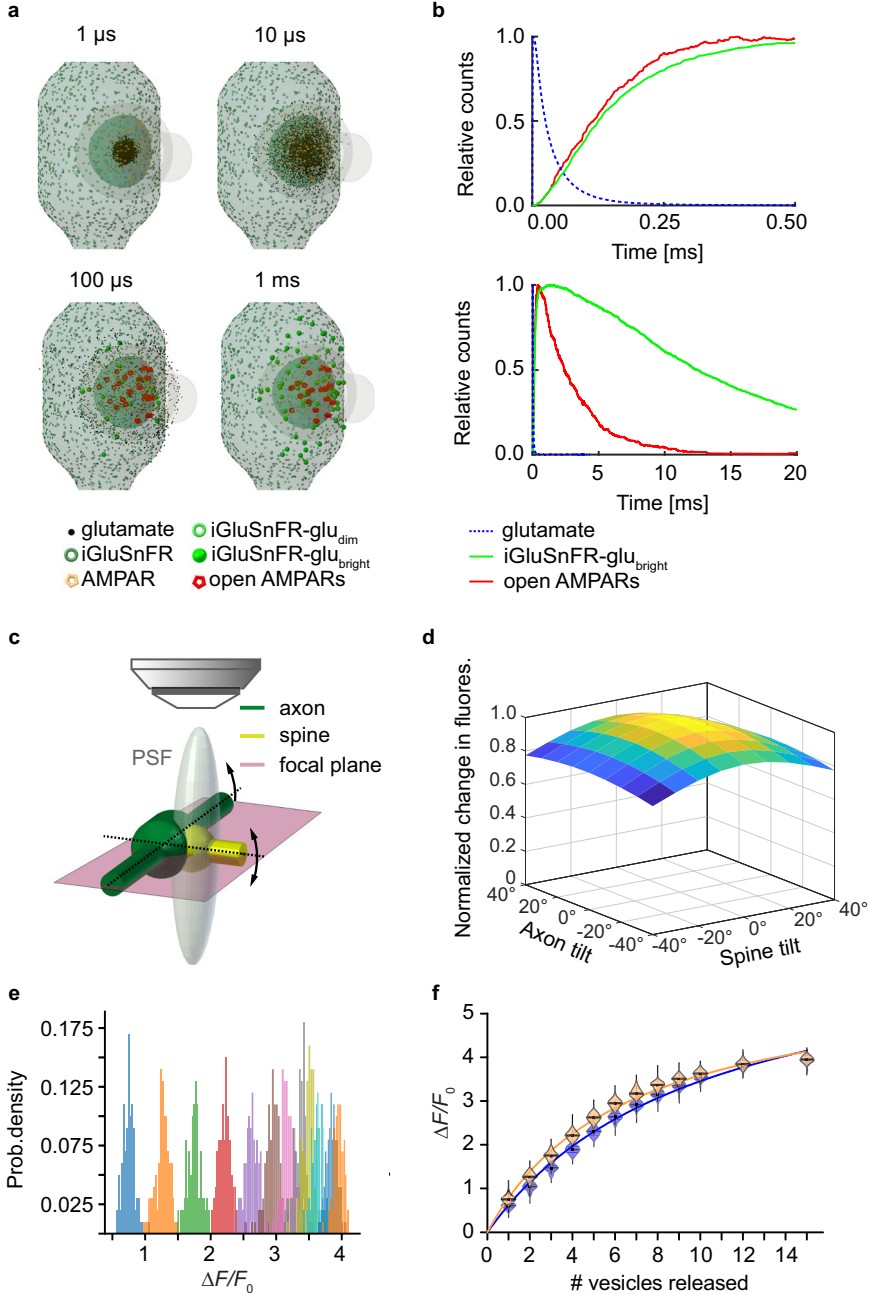

**Fig. 5 | Monte-Carlo-simulation of glutamate diffusion in the synaptic cleft.**
**a** Simulated glutamate dynamics after release of a single vesicle. The model consists of a presynaptic terminal with iGluSnFR molecules opposed to a spine with AMPARs randomly distributed in a disc of 300 nm, separated by a 20-nm synaptic cleft. The synapse is embedded in a network of astrocytes equipped with glutamate transporters (not shown). **b** Kinetics of AMPAR opening and iGluSnFR entering the bright fluorescent state upon release of a single vesicle containing 3000 molecules of glutamate (upper panel) and expanded time scale (lower panel). **c** Model geometry including the point spread function (PSF) of the microscope. Different orientations of axon and spine with respect to the optical axis were evaluated. **d** Tilting the axon or the spine reduces $\Delta F/F_0$, as more iGluSnFR molecules outside the synaptic cleft fall within the PSF, contributing to the resting fluorescence $F_0$. **e** Histogram of peak iGluSnFR ($\Delta F/F_0$) after simultaneous release of a specific number of vesicles (1–15 vesicles, 100 runs each, 3000 glutamate molecules per vesicle). **f** Summary of release simulations at optimal cleft orientation (orange violins) and with 40° spine tilt (blue violins). Curves are hyperbolic functions (least squares fit). Source data are provided as a Source Data file.

histograms (Supplementary Fig. 8) and to the exact saturation value (Supplementary Fig. 9), which we expect to be slightly higher in tilted synapses (Fig. 5d). In our sample of 27 boutons (ref. 32), the estimated number of docked vesicles ranged from 1 to 8, with only 2 boutons having just one docked vesicle (Fig. 6d). Changing $[Ca^{2+}]_e$ from 1 to 4 mM increased $p_{ves}$ 9.9-fold (median change; Fig. 6d).

We estimated the iGluSnFR response to the release of a single vesicle ($q$) with three different approaches of increasing complexity: (1) by analyzing the amplitude of successes under low release probability conditions, (2) by measuring the amplitude of desynchronized events during $[Sr^{2+}]_e$ replacement experiments, and (3) by fitting a binomial model to the complete distribution of successes and failures from a single bouton (Fig. 6e). The resulting estimates of $q$ were very consistent between methods, suggesting that our fitting procedure correctly extracts the presynaptic quantal size. It is important to note that the absolute amplitude ($\Delta F/F_0$) of $q$ depends not only on the indicator, but also on the spatial and temporal resolution of the microscope as it is trying to catch

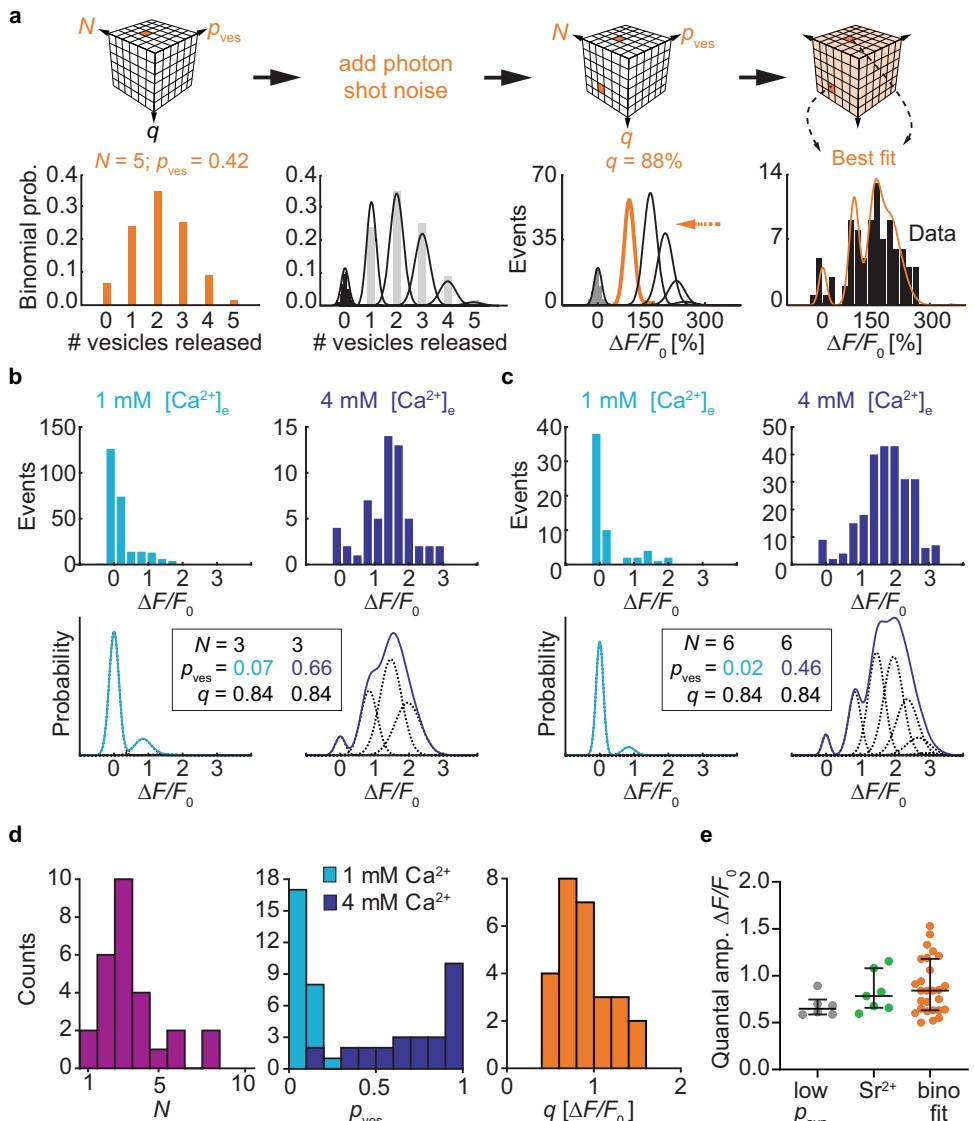

**Fig. 6 | Extracting $N$, $p_{ves}$, and $q$ of Schaffer collateral synapses using a binomial model. a** Extracting quantal parameters by exhaustive search. For every combination of $N$ and $p_{ves}$, the binomial probabilities of the possible outcomes were calculated (here: $N = 5$ vesicles and $p_{ves} = 0.42$). Without stimulation, amplitudes were normally distributed around zero (black histogram). The width of this noise distribution was used to determine the width of the expected Gaussian probability density functions. A chosen quantal amplitude $q$ (here: 88% $\Delta F/F_0$) and iGluSnFR saturation function ($B_{max} = 6$ $\Delta F/F_0$, orange arrow) sets the spacing of the Gaussians. The resulting probability density function (sum of the Gaussians) was compared to the measured amplitude distribution (black bars, recorded in 2 mM [Ca$^{2+}$]$_e$). The root mean square (RMS) error was calculated and the best fit (shown here) was selected. **b**, **c** Single bouton response distributions recorded in 1 mM [Ca$^{2+}$]$_e$ and 4 mM [Ca$^{2+}$]$_e$. Two examples from two different slice cultures. The binomial fitting procedure was applied to both histograms, searching for the best

combined fit under the condition that $N$ and $q$ had to be identical in 1 mM [Ca$^{2+}$]$_e$ and 4 mM [Ca$^{2+}$]$_e$ while $p_{ves}$ could be different. Best fit probability density functions and extracted parameters are shown below the experimental data. **d** Extracted quantal parameters, $N$ median: 3 vesicles, IQR: 2–4 vesicles; $p_{ves}$ in 1 mM [Ca$^{2+}$]$_e$ median: 0.08, IQR: 0.04–0.12; $p_{ves}$ in 4 mM [Ca$^{2+}$]$_e$ median: 0.79, IQR = 0.55–0.94; $q$ median = 0.84 $\Delta F/F_0$, IQR: 0.63–1.18 $\Delta F/F_0$, $n = 27$ boutons. **e** Estimating quantal amplitude by three different approaches using independent datasets. Comparing the three approaches, the estimated values of $q$ were not significantly different (Kruskal–Wallis test, $p = 0.22$). $p_{syn} < 0.5$ boutons in 2 mM [Ca$^{2+}$]$_e$ median: 0.65 $\Delta F/F_0$, IQR: 0.59–0.75 $\Delta F/F_0$, $n = 6$ boutons; delayed events in 4 mM [Sr$^{2+}$]$_e$ wash-in experiments median: 0.78 $\Delta F/F_0$, IQR: 0.66–1.08 $\Delta F/F_0$, $n = 7$ boutons; binomial fitting procedure median: 0.84 $\Delta F/F_0$, IQR: 0.63–1.18 $\Delta F/F_0$, $n = 27$ boutons. Values are given as median with IQR. Source data are provided as a Source Data file.

the peak fluorescence caused by a rapidly diffusing cloud of glutamate.

## Vesicular release probability predicts synaptic strength in low Ca$^{2+}$

Lastly, we used our dataset to determine which synaptic parameter has the strongest impact on (pre-)synaptic strength. For each synapse, we calculated its synaptic strength as the product of quantal parameters ($p_{ves} \times N \times q$) under low and under high [Ca$^{2+}$]$_e$ conditions. Compared to simply averaging all iGluSnFR responses, this calculation removed

the compressive effect of indicator saturation. In 1 mM [Ca$^{2+}$]$_e$, $p_{ves}$ was strongly correlated with synaptic strength ($R^2 = 0.85$, Fig. 7a), but the number of readily releasable vesicles ($N$) was not. In 4 mM [Ca$^{2+}$]$_e$, $N$ ($R^2 = 0.28$) and $p_{ves}$ ($R^2 = 0.42$) jointly determined synaptic strength, suggesting that the number of readily releasable vesicles limits the strength of a synapse under conditions of high release probability (Fig. 7b). In both conditions, quantal size $q$ had little impact on synaptic strength. Non-parametric tests between all measured parameters confirmed those findings (Supplementary Fig. 10) and in addition, revealed that weak synapses were strongly facilitating when stimulated

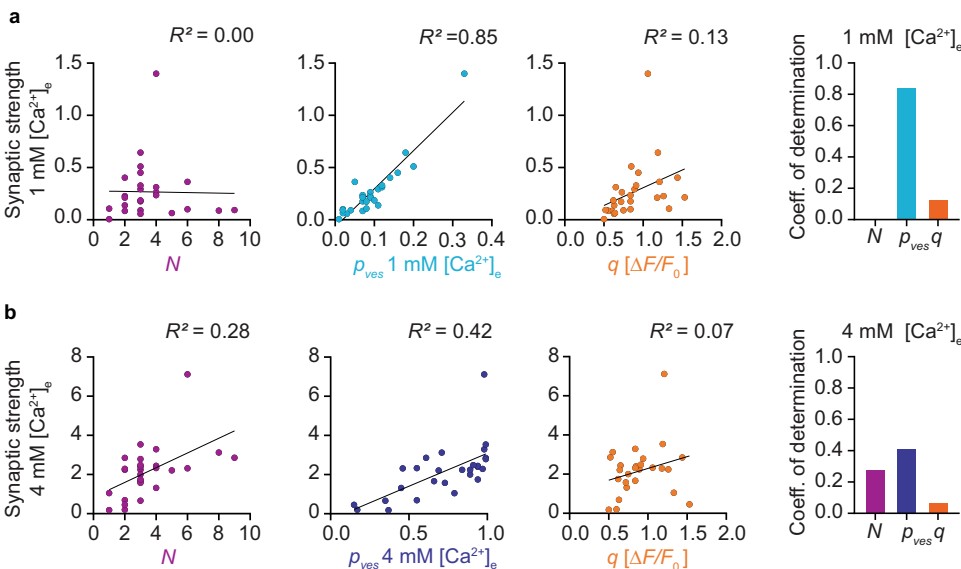

**Fig. 7 | Quantal parameters and synaptic strength. a** Linear correlations between the extracted quantal parameters $N$, $p_{ves}$, and $q$ and the calculated synaptic strength in 1 mM $[Ca^{2+}]_e$ ($n = 27$ boutons). **b** Linear correlations between quantal parameters and synaptic strength in 4 mM $[Ca^{2+}]_e$ ($n = 27$ boutons). Source data are provided as a Source Data file.

by a pair of pulses in high $[Ca^{2+}]_e$ (Spearman's $r = -0.76$). This was expected, as high $p_{ves}$ boutons in high $[Ca^{2+}]_e$ spend a sizable fraction of their releasable vesicles on the first pulse and tend to display paired-pulse depression (Fig. 4f).

In summary, most Schaffer collateral synapses in our sample were capable of increasing the glutamate output per AP under high release probability conditions and produced amplitude distributions consistent with binomial statistics. The vesicular release probability was remarkably variable between individual presynaptic terminals, suggesting that the number of calcium channels and/or their distribution with respect to the releasable vesicles is not uniform[33]. Whether Schaffer collateral synapses operate in a high or low release probability regime when the animal is engaged in a behavioral task and neuromodulatory inputs are active remains to be determined.

## Discussion

### Different approaches to optical quantal analysis
By measuring cleft glutamate transients at Schaffer collateral synapses under conditions of low and high release probability, we directly show the impressive dynamic range of individual boutons. The capacity for multivesicular release has initially been inferred from the analysis of postsynaptic currents at various synapses in the CNS[21]. Optical approaches to quantal analysis were based on the analysis of spine $Ca^{2+}$ transients (EPSCaTs) as a proxy for postsynaptic depolarization[19,34,35]. Compared to EPSCaT measurements, glutamate imaging has four distinct advantages: First, the unitary response to the release of a single vesicle (quantal size $q$, here measured in units of $\Delta F/F_O$) is largely determined by the properties of iGluSnFR and the resolution of the microscope and thus very similar across individual synapses. EPSCaT amplitude, in contrast, depends on the density of NMDARs, AMPARs, and the diluting volume of the spine. The unitary EPSCaT is therefore different in every synapse, and it is practically impossible to wait for spontaneous EPSCaTs (the equivalent of "miniature end-plate potentials") while imaging continuously. This is a serious drawback, as knowing $q$ is at the heart of true quantal analysis. Second, EPSCaTs are mediated by voltage-dependent NMDA receptors and voltage-gated calcium channels. Dendritic depolarization by other active synapses can therefore influence EPSCaT amplitude at the synapse under scrutiny. iGluSnFR signals, in contrast, are highly localized and unlikely to be contaminated by the activity of nearby synapses. Third, EPSCaTs are

sensitive to the extracellular $Ca^{2+}$ concentration while iGluSnFR is not. The calcium-independent read-out made it possible for us to directly investigate the impact of $[Ca^{2+}]_e$ on the release machinery and to replace $Ca^{2+}$ by $Sr^{2+}$, generating desynchronized fusion events. Fourth, iGluSnFR directly probes the presynaptic function from a neuron of known identity. Several boutons on the same axon can be probed in parallel or sequentially, removing the bias towards strongly responding synapses that troubles EPSCaT analysis. Although other optical methods, such as styryl dyes or pHluorin-based indicators, share some of the advantages of iGluSnFR, they report vesicle fusion and not glutamate release.

The increase in synaptic glutamate associated with increased release probability observed in previous studies has been challenged by alternative explanations, namely diffusion of glutamate from adjacent sites[36]. Could the larger glutamate concentrations we observed under high release probability conditions be a result of synaptic spill-over? In contrast to extracellular field stimulation which activates an unknown number of axons, we triggered APs in a single CA3 pyramidal cell. As presynaptic boutons on a single active axon are spatially well separated, spillover of glutamate is extremely unlikely. As a precaution, we performed all imaging experiments at 33 °C to ensure efficient glutamate uptake by astrocytes. Under both low and high release probability conditions, iGluSnFR transients peaked in the same region of the bouton, further indicating that the source of glutamate was solely the active zone of the bouton under investigation.

### Saturation of glutamate sensors in the synaptic cleft
The steady-state glutamate affinity of iGluSnFR ($K_d = 2.1\,\mu M$ on neurons[14]) is similar to neuronal AMPARs ($3–10\,\mu M$[37]). Glutamate concentrations are thought to reach $1.1\,mM$ in hippocampal synapses[38], but AMPAR occupancy is nevertheless quite low[39]. As the glutamate transient in the synaptic cleft is very short[40], the *on*-rate rather than the steady-state affinity determines the occupancy after vesicular glutamate release. iGluSnFR may be a good proxy for the activation of AMPARs (Fig. 4), but it will not linearly report cleft glutamate. To extract quantal parameters from iGluSnFR signals, it is essential to correct for indicator non-linearity, as quantal peaks will not appear at integer multiples of $q$. Our simulations of multivesicular release (Fig. 5f) were well fit by a hyperbolic curve, but the asymptotic value was 600% (700% for tilted synapses), higher than iGluSnFR's

maximum $\Delta F/F_O$ of 440%. This apparent paradox is caused by extra-synaptic iGluSnFR molecules that contribute to $F_O$, but never experience high glutamate concentrations. As a consequence, the signals of 1–10 simultaneously released vesicles seem to be far from saturation. How much do our results depend on the exact saturation value? We run the entire parameter extraction procedure with three different saturation values (440%, 600%, 700%, Supplementary Fig. 9). Within the range caused by different tilt of synapses (600–700%) extracted parameters were very similar. Using the literature values for iGluSnFR (440%) for bouton saturation was not compatible with the results of our Monte-Carlo modeling (Fig. 5f).

As iGluSnFR competes with excitatory amino acids transporters (EAATs) for glutamate, it slows down the clearance of bulk glutamate from the extracellular space[41]. However, diffusion of glutamate out of the synaptic cleft occurs at much shorter time scales (<100 μs) than uptake by EAATs. Our measurements of synaptic strength between pairs of connected CA3-CA1 neurons showed no significant difference between iGluSnFR-expressing and non-transfected CA3 neurons (Supplementary Fig. 7), demonstating that buffering effects of iGluSnFR did not affect postsynaptic AMPAR currents. The situation might be different for global expression of iGluSnFR[42] or during periods of dense neuronal activity.

### Quantal parameters and their variability

Our estimate of 1–9 readily releasable SVs (median = 3) is in line with other functional measurements based on statistic of synaptic transmission to quantify the number of release sites[19,43]. Electron tomography of glutamatergic synapses from rapidly frozen organotypic hippocampal slice cultures showed 10–12 docked vesicles per active zone[44], but not all docked vesicles may be release-competent. A recent study of primary hippocampal cultures determined the number of distinct release sites per active zone through a clustering method[7]. They estimated 10 release sites per active zone (assuming a release site diameter of 70 nm), but it is not clear whether all of these release sites are constantly occupied by release-ready vesicles. Another study on dissociated neurons, using total internal reflection fluorescence microscopy (TIRF) to monitor vesicle release with pHluorin, estimated 3–8 release sites per active zone[45]. Thus, while the absolute number of release-ready vesicles may vary with the preparation and age of the culture[46], the result of our analysis is consistent with estimates from other functional imaging approaches.

Quantal size, the iGluSnFR fluorescence change in response to the release of a single vesicle ($q$) varied between boutons (CV = 0.33), with an average amplitude very similar to the desynchonized events we observed during $Sr^{2+}$ wash-in. This variability is comparable to EM measurements of SV diameter, where the mean vesicle volume varied between individual CA1 synapses up to 5-fold (5000 nm³ to 25,000 nm³) with a CV of 0.35 (Refs. [47], [48]). There is also variability of vesicle volume within boutons. Based on EM vesicle diameter measurements[47], the distribution of vesicle volumes is thought to be skewed with a CV = 0.38 (Ref. [49]). The fact that we and others[50] observed a pronounced gap in the iGluSnFR response histogram between single- and multivesicular events is not consistent with such a large variability. In our fitting procedure to determine the quantal parameters, we did not account for any variability in glutamate content. The only source of variability we included was photon shot noise, which we could precisely determine for every bouton from the fluctuations of baseline fluorescence. Including a term for quantal variability led to unrealistically broad distributions that were not compatible with the multi-peaked histograms we measured in high $Ca^{2+}$. Thus, it is possible that only vesicles with a specific diameter or filling state can dock and fuse, which would be a novel quality control mechanism ensuring quantal uniformity[9]. In summary, our functional measurements from live synapses suggest that glutamate quanta within a synapse are more uniform than ultrastructural diameter measurements would suggest.

### The importance of $p_{ves}$ and $N$ for short-term plasticity depends on the state of the synapse

Our results show that even the smallest synapses in the brain are capable of multivesicular release MVR. However, in 1 mM $[Ca^{2+}]_e$, which is close to physiological $[Ca^{2+}]_e$ in awake animals[13], $p_{ves}$ is low (0.01–0.23) and multivesicular release events are quite rare. We show that under these conditions, $p_{ves}$ is highly variable between individual boutons and is therefore the main determinant of synaptic strength. Under high $p_{syn}$ conditions, $N$ also becomes important as it limits synaptic output in multivesicular mode. In vivo, the relative impact of $p_{ves}$ and $N$ on synaptic output may vary depending on the neuromodulatory state and the frequency of transmission, and the fraction of multivesicular release events will be variable, too.

The use of available resources (the readily releasable pool of vesicles) is expected to affect the short-term plasticity of a synapse[3,25]. We found that Schaffer collateral synapses that released little glutamate in response to a single AP produced pronounced paired-pulse facilitation in 4 mM $[Ca^{2+}]_e$ (Supplementary Fig. 10), consistent with a previous glutamate imaging study on cultured hippocampal neurons[51]. A recent study[12] combined glutamate and calcium imaging at individual boutons in 2 mM $[Ca^{2+}]_e$ and found short-term plasticity to be independent of bouton $[Ca^{2+}]_i$. Similarly, we found that in 1 mM $[Ca^{2+}]_e$, correlations between PPR and quantal parameters were insignificant (Supplementary Fig. 10), although PPR at individual synapses was quite variable under these conditions (Fig. 5f). We conclude that short-term plasticity and differential filtering of high frequency signals as described by Markram and Tsodyks[52] can be readily linked to quantal parameters in high $Ca^{2+}$, but may be dominated by other factors under more physiological, low-use conditions. Proximity to astrocytic processes may play a role[53] as well as the age of the synapse, as mature synapses are able to use their vesicle complement much more efficiently[46]. In the future, combining chronic imaging with optical quantal analysis would be an elegant approach to investigate the dynamics of quantal parameters over the lifetime of a synapse.

## Methods

### Slice culture preparation

Organotypic hippocampal slices were prepared from Wistar rats of either sex at postnatal day 5–7. Briefly, dissected hippocampi[54] were cut into 400 μm slices with a tissue chopper and placed on a porous membrane (Millicell CM, Millipore). Cultures were maintained at 37 °C, 5% $CO_2$ in a medium containing (for 500 ml): 394 ml Minimal Essential Medium (Sigma M7278), 100 ml heat inactivated donor horse serum (H1138 Sigma), 1 mM L-glutamine (Gibco 25030-024), 0.01 mg ml⁻¹ insulin (Sigma I6634), 1.45 ml 5 M NaCl (S5150 Sigma), 2 mM $MgSO_4$ (Fluka 63126), 1.44 mM $CaCl_2$ (Fluka 21114), 0.00125% ascorbic acid (Fluka 11140), 13 mM D-glucose (Fluka 49152). No antibiotics were added to the culture medium. The medium was partially exchanged (60–70%) twice per week. Wistar rats were housed and bred at the University Medical Center Hamburg-Eppendorf (UKE). All procedures were performed in compliance with German law (Tierschutzgesetz der Bundesrepublik Deutschland, TierSchG) and according to the guidelines of Directive 2010/63/EU. Protocols were approved by the Behörde für Justiz und Verbraucherschutz (BJV) - Lebensmittelsicherheit und Veterinärwesen, Hamburg.

### Plasmids and electroporation procedure

iGluSnFR, a gift from Loren Looger (Addgene plasmid #41732) and tdimer2[55] were each subcloned into a neuron-specific expression vector (pCI) under the control of the human synapsin1 promoter. Plasmids were diluted to 20 ng/μL and 40 ng/μL for tdimer2 and iGluSnFR, respectively, in K-gluconate-based solution consisting of (in mM): 135 K-gluconate, 4 $MgCl_2$, 4 $Na_2$-ATP, 0.4 Na-GTP, 10 $Na_2$-phosphocreatine, 3 ascorbate and 10 HEPES (pH 7.2). CA3 pyramidal neurons were transfected by single-cell electroporation between DIV 17 and DIV 25

with a mixture of the two plasmids. During the electroporation procedure, slice cultures were maintained in pre-warmed HEPES-buffered solution consisting of (in mM): 145 NaCl, 10 HEPES, 25 D-glucose, 1 MgCl$_2$ and 2 CaCl$_2$ (pH 7.4, sterile filtered). An Axoporator 800 A (Molecular Devices) was used to deliver 50 voltage pulses (−12 V, 0.5 ms) at 50 Hz[56].

## Solutions and electrophysiology

Experiments were performed 2–4 days after electroporation. Hippocampal slice cultures were placed in the recording chamber of the microscope and superfused with artificial cerebrospinal fluid (ACSF) containing (in mM): 127 NaCl, 25 NaHCO$_3$, 25 D-glucose, 1.25 NaH$_2$PO$_4$, 2.5 KCl, 2 CaCl$_2$, 1 MgCl$_2$. ACSF was saturated with 95% O$_2$ and 5% CO$_2$. In the experiments where [Ca$^{2+}$]$_e$ was changed, we switched from 1 mM Ca$^{2+}$, 4 mM Mg$^{2+}$ to 4 mM Ca$^{2+}$, 1 mM Mg$^{2+}$ to keep the divalent ion concentration constant. Patch pipettes with a tip resistance of 3.5 to 4.5 MΩ were filled with (in mM): 135 K-gluconate, 4 MgCl$_2$, 4 Na$_2$-ATP, 0.4 Na-GTP, 10 Na$_2$-phosphocreatine, 3 ascorbate and 10 HEPES (pH 7.2). Experiments were performed at 33 °C ± 1 °C by controlling the temperature of the ACSF with an in-line heating system and heating the oil immersion condenser with a Peltier element. Whole-cell recordings from transfected CA3 pyramidal neurons were made with a Multiclamp 700B amplifier (Molecular Devices) under the control of Ephus software written in Matlab[57]. CA3 neurons were held in current clamp and stimulated through the patch pipette by brief electrical pulses (2–3 ms, 1.5–3.5 nA) to induce single APs. Individual trials (single pulse or paired-pulse) were delivered at a frequency of 0.1 Hz. The analog signals were filtered at 6 kHz and digitized at 10 kHz. For dual patch experiments, CA1 neurons were recorded in voltage clamp. Access resistance ($R_{acc}$) was monitored continuously throughout the experiment and recordings with $R_{acc} > 20$ MΩ were discarded. To isolate AMPA receptor responses, 10 μM CPP-ene was added to the perfusate.

For extracellular synaptic stimulation, a monopolar electrode was placed in stratum radiatum and two 0.2 ms pulses, 48 ms apart, were delivered using an ISO-Flex stimulator (A.M.P.I.). Stimulation intensity was adjusted to be subthreshold for APs. 10 mM γ-DGG was added to the bath in experiments were the fold change in AMPARs was probed under decrease of AMPARs saturation.

## Two-photon microscopy

The custom-built two-photon imaging setup was based on an Olympus BX51WI microscope controlled by a customized version the open-source software package ScanImage[58] written in MATLAB (Math-Works). We used a pulsed Ti:Sapphire laser (MaiTai DeepSee, Spectra Physics) tuned to 980 nm to simultaneously excite both the cytoplasmic tdimer2 and the membrane bound iGluSnFR. Red and green fluorescence was detected through the objective (LUMPLFLN 60XW, ×60, 1.0 NA, Olympus) and through the oil immersion condenser (1.4 NA, Olympus) using 2 pairs of photomultiplier tubes (PMTs, H7422P-40SEL, Hamamatsu), 560 DXCR dichroic mirrors and 525/50 and 607/70 emission filters (Chroma Technology) were used to separate green and red fluorescence. Excitation light was blocked by short-pass filters (ET700SP-2P, Chroma). ScanImage was modified to allow arbitrary line scanning. To measure iGluSnFR signals with a high signal-to-noise ratio, spiral scans were acquired to sample the surface of individual boutons. For single pulse stimulation, we acquired 44 spiral lines at 500 Hz or 330 Hz. For paired-pulse pulse stimulation (48 ms ISI), we acquired 64 spiral lines at 500 Hz. Photomultiplier dark noise was measured before shutter opening and subtracted for every trial.

## Drift correction

To compensate for movements of the tissue during long imaging sessions, we developed an automated drift correction algorithm to re-center the synapse of interest. As spatial reference, we obtained a series of optical sections (z-step size: 0.5 μm) that were interpolated to

0.25 μm. For drift correction, we acquired a single frame-scan (test image) and performed subpixel image registration against the stack of reference images to extract lateral drift. In a second step, the overlapping regions from both, the test image and reference images were compared via cross correlation to reveal axial drift. Drift was compensated by adding offsets to the xy-scanner command voltages and by moving the objective to the correct z-position. Drift correction was typically completed within 0.3 s and performed before each stimulation trial.

## Analysis of fluorescence transients

In case of a release event ('success'), a spiral scan covering the entire bouton may hit the diffusing cloud of glutamate just once or several times per line. We had no prior knowledge about the precise location of fusion events on the bouton surface. To maximize the signal-to-noise ratio in every trial, we assigned a dynamic region of interest (ROI): Pixel columns (i.e. spatial positions) were sorted according to the change in fluorescence ($\Delta F$) in each column. In 'success' trials (average $\Delta F > 2\sigma$ of baseline noise), only columns which displayed a clear change in fluorescence ($\Delta F > \frac{1}{2}$ max ($\Delta F$)) were evaluated. In 'failure' trials (when $\Delta F$ of each column of the ROI was 5% > than $\Delta F$ of the corresponding columns in the baseline), the columns selected in the last 'success' trial were evaluated. At that stage, the classification used for ROI positioning (success vs. failure) was preliminary. Indeed, some 'failure' trials did show small fluorescent transients in the more sensitive ROI-based analysis. Boutons with a full width at half maximum (FWHM) of the amplitude distribution of the baseline (i.e. non-stimulated trials) >0.4 were rejected as the imaging conditions were considered non-optimal and not considered for further analysis (Supplementary Fig. 2). To correct for bleaching, we fit an exponential decay to $F_0$ in failure trials. We corrected all data for bleaching by subtracting monoexponential fits acquired from the average fluorescence time course of failures. This bleach time constant was used to establish a photobleaching correction for each trial. To measure the amplitude iGluSnFR changes in fluorescence and to distinguish successful release of glutamate from failures, we used a template-based fitting algorithm. For each bouton we extracted a characteristic decay time constant by fitting a mono-exponential function to the average bleach-corrected iGluSnFR signals. To estimate the glutamate transient amplitude for every trial we kept the amplitude as the only free parameter. To check for overall stability, we measured the mean $F_0$ (Baseline iGluSnFR signal) in each trial. If a trial had a $F_0 > 2\sigma$ measured out of the average $F_0$ of all trials then that trial was removed from the analysis.

The number of successes divided by the number of trials is the probability of release for a synapse ($p_{syn}$). We define the success amplitude to a single stimulus as the cleft glutamate ([Glu]$_{success}$).

## Synapse modeling and glutamate release simulation

Release of glutamate and the time profile of iGluSnFR fluorescence were simulated using a Monte Carlo method (MCell) that considers the stochastic nature of molecule diffusion as well as that of reaction between molecules. The model consisted of an axon (diameter 0.2 μm, length 3 μm) with a varicosity representing the bouton (diameter 0.5 μm, length 0.5 μm), a hemispheric structure representing the spine (diameter 0.4 μm) attached to a cylindrical spine neck (diameter 0.2 μm). Active zone and postsynaptic density were defined as circular areas (diameter 300 nm) separated by the synaptic cleft (20 nm)[59]. Axon and spine were enclosed by an astrocytic compartment (width of extracellular space: 20 nm). Boundary conditions for the entire system were reflective. Glutamate transporters (GluT) were placed on astrocytic membranes at a density of 10.000 μm$^{-2}$. AMPARs were restricted to the PSD at a density of 1.200 μm$^{-2}$ (resulting in ~85 receptors at a PSD diameter of 300 nm). Vesicle fusion was modeled by an instantaneous injection of glutamate at a fixed position (center of the active zone).

The glutamate content of a single vesicle was estimated to be in the range of 2.000–3.000 molecules[27]. The diffusion coefficient of glutamate[60] was set to 200 $\mu m^2/s$. To study the consequences of univesicular and multivesicular release, we varied the number of released glutamate molecules between 3000 and 45.000 (1–15 vesicles).

Our model of iGluSnFR is based on a two-step reaction where rapid binding of glutamate to iGluSnFR is followed by a slower conformational change with concomitant increase in fluorescence:

$$Glu + iGluSnFR \underset{k_{-1}}{\overset{k_{+1}}{\rightleftharpoons}} Glu - iGluSnFR_{dim} \underset{k_{-2}}{\overset{k_{+2}}{\rightleftharpoons}} Glu - iGluSnFR_{bright} \quad (2)$$

However, free glutamate dissipates from the synaptic cleft in ~100 $\mu s$, preventing equilibration of the first reaction. Therefore, the reaction can be linearized to

$$Glu + iGluSnFR \overset{k_{+1}}{\rightarrow} Glu - iGluSnFR_{dim} \overset{k_{+2}}{\rightarrow} Glu - iGluSnFR_{bright} \overset{k_{-2}}{\rightarrow} Glu + iGluSnFR_{dim}$$
$$(3)$$

Association and dissociation kinetics were measured in vitro revealing that the fluorescence change occurs in the second, isomerisation step[26]. Kinetic measurements of the conformational change of the purified protein gave $k_{+2}$ of 569 s$^{-1}$ and $k_{-2}$ of 110 s$^{-1}$ (ref. [49]). The equivalent values from our cell-based measurements were 2481 s$^{-1}$ and 111 s$^{-1}$, respectively (Supplementary Fig. 11). Such differences are expected between purified protein and cell-based measurements[10]. In addition to iGluSnFR, we included a kinetic AMPA receptor model[29] based on recordings (22–24 °C) from rat hippocampal slices (age: p15–p24). This model represents GluA1/GluA2 heteromers that are typical for CA3-CA1 synapses. We adjusted the rate constants to match the temperature of our experiments (33 °C).

### Quantal analysis
We wrote custom code (Matlab) to extract quantal parameters (https://github.com/toertner/optical-quantal-analysis). Analysis of optical signals is fundamentally different from EPSP analysis as the sources of noise are different and indicator saturation has to be considered. When collecting (green) fluorescence, the photon 'shot noise' follows Poisson statistics. For each bouton, we measured the standard deviation of baseline fluorescence ($\sigma$) in each trial before stimulation (baseline noise). Imaging conditions varied between individual experiments (depth in tissue, expression level, laser power) and we discarded experiments with baseline noise above 40% (Supplementary Fig. 2). For the remaining boutons, we generated predicted amplitude distributions based on binomial statistics and hyperbolic correction for non-linearity. The width of the success distributions (Gaussians) was determined by the expected photon shot noise calculated from the baseline noise. Propagation of the shot noise was considered in quadrature:

$$\sigma Q = \sqrt{\sigma a^2 + \sigma a^2} \quad (4)$$

where $a$ represents a probability density function for $N = 1$ and $Q$ represents the propagated error. We explored the following parameter space: $N$: 1 to 15; $p_{ves}$: 0.1 to 0.99 (0.01 steps); $q$: 0.5 to 2.0 $\Delta F/F_0$ (0.01 steps). The full set of quantal analyses is available at Zenodo (https://doi.org/10.5281/zenodo.6997652).

### Statistical analysis
Normality was tested using D'Agostino-Pearson omnibus normality test (GraphPad Prism 8). To test for significant differences between population means, paired t test (two-sided) or the non-parametric Wilcoxon-Signed rank test. To compare independent populations, we used the unpaired t test (two-sided) or the non-parametric Mann–Whitney test as appropriate. To test for significant correlation between populations we used the non-parametric Spearman correlation (two-sided) test. Statistical significance was assumed when $p \leq 0.05$. Symbols used for assigning significance in figures: not significant (n.s.), $p > 0.05$; significant, $p \leq 0.05$(*), $p \leq 0.01$(**), $p \leq 0.001$(***) and $p \leq 0.0001$(****).

### Reporting summary
Further information on research design is available in the Nature Research Reporting Summary linked to this article.

## Data availability
The data generated in this study have been deposited in the Zenodo database under accession code 6997652. Source data are provided with this paper.

## Code availability
Parameter files used for the MCell simulation (https://github.com/toertner/iGluSnFR-simulation) and Matlab functions for quantal analysis (https://github.com/toertner/optical-quantal-analysis) have been deposited on GitHub.

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

## Acknowledgements

We thank Iris Ohmert and Sabine Graf for slice culture preparation and excellent technical support. We thank Gary S. Bhumbra for helpful advice about quantal parameter extraction, Christine E. Gee and Mauro Pulin for critical reading of the manuscript. This study was supported by the German Research Foundation DFG (SPP 1665 (322093511, to T.G.O); SPP 1926 (315380903, to J.S.W.); SFB 936 (178316478, to T.G.O. and J.S.W.); FOR 2419 (278170285, to T.G.O. and J.S.W.)), Landesforschungsförderung Hamburg Z-FR LF (to T.G.O.); BBSRC project grant BB/S003894/1 (to K.T.), and the British Heart Foundation Intermediate Basic Science Research Fellowship FS/17/56/32925 (to N.H.).

## Author contributions

C.D.D., J.S.W., and T.G.O. designed the experiments. C.D.D. and T.G.O. prepared the manuscript. C.D.D. performed synaptic imaging experiments and analyzed the data. C.S. performed Monte-Carlo simulations and wrote microscope control software. C.D.D., T.G.O., and C.S. wrote analysis software. N.H. and K.T. provided kinetic measurements of iGluSnFR. All authors read and approved the manuscript.

## Funding

## Competing interests

The authors declare no competing interests.
