## [Peer Review File · Nature Communications]

Vesicular release probability sets the strength of individual Schaffer collateral synapsesReviewers' comments:

Reviewer #1 (Remarks to the Author):

A quantitative description of neurotransmitter release was originally developed by Katz using the frog neuromuscular junction as a model system, in the 1950s. Since then, a number of groups have estimated the quantal parameters N , Pr/P_v , and q at peripheral and central synapses. As the authors mention in their introduction, there are a number of limitations to keep in mind when attempting to derive these quantal parameters at small central synapses, which can be largely attributed to electrotonic filtering of synaptic responses arising at a distance from the recording site (typically the soma). In this work, the authors develop a new experimental approach to estimate P_v at Schaffer collateral synapses in organotypic slice cultures. The approach is based on the combined use of optical reporters of extracellular glutamate (iGluSnFR) and sophisticated high frequency spiral scan two-photon imaging. The main findings of this work indicate that Schaffer collateral synapses switch from a mostly univesicular to multivesicular release at increasing extracellular calcium concentrations. A similar conclusion was reached by Christie and Jahr (J Neurosci 2006), but this work was solely based on the use of competitive low affinity antagonists for AMPA receptors. For this reason, one could claim that the data from Christie and Jahr provided only an indirect proof of what the current paper shows with much more direct evidence. The work is consistent with that of Jensen et al (Nat Comm 2019), in which the authors imaged iGluSnFR and Cal-590 at individual boutons. Overall, the manuscript is clearly presented, the imaging approaches are elegant and rigorous, and the data interpretation is fair. Where the paper falls short, in my opinion, is novelty, and I hope the points raised below will ultimately help to overcome this potential limitation.

It seems like the quantal amplitude determined using the $Sr+$ approach was only recorded in $n=3$ boutons. The authors use three methods to measure q , but one is still left to wonder whether the conclusion that bouton-to-bouton variability in q is a side effect of the low sensitivity of the detection approaches. In other words, if an average synaptic vesicle contains 2,000 glutamate molecules, what is the smallest change in the number of these molecules that can be detected by the three approaches described on page 16? Based on these considerations, one cannot exclude the possibility that relying on such a small sample size might introduce errors for the quantal analysis presented in this manuscript. Increasing the sample size would help.

The authors state that a typical bouton (assuming a typical bouton truly exists) displays multivesicular release 4% of times at $[Ca^{2+}]_e=1$ mM, and that this proportion increases to 77% when $[Ca^{2+}]_e=4$ mM. The physiological concentration of calcium in the CSF is thought to be 1.2 mM, and fluctuations in this baseline level can occur as a result of neuronal activity, etc. It would be valuable to take advantage of the model to show how the probability of uni/multivesicular release change over $[Ca^{2+}]_e$ ranging from 0-4 mM. How steep is this relationship? Is such a high proportion of multivesicular release (77%) ever reached under physiological conditions, where $[Ca^{2+}]_e=4$ mM is unlikely to be reached?

The authors conclude that P_{syn} is variable among terminals, and suggest that this could be attributed to differences in the distance between the RRP and calcium channels. This explanation seems a little simplistic, as one would imagine that differences in the number of calcium channels (not just the distance between the RRP and the calcium channels) could also contribute to this effect. Can the authors explain more carefully their reasoning? Are they implying that the RRP and the calcium channels must be separate from each other (e.g. Nadkarni et al, PNAS 2012) or that the calcium channels can be sparse within the domain of the active zone where the synaptic vesicles are docked (e.g. Scimemi and Diamond, 2012)?

The finding that there is variability in P_{syn} from bouton to bouton is not particularly unexpected, but how does the magnitude of this variability change at varying $[Ca^{2+}]_e$? How does $[Ca^{2+}]_e$ affect the variability of P_v within a given bouton?

How stable are the properties of these synapses over time? Clearly, calcium channels have a limited

lifetime in the presynaptic active zone, and the authors have an impressive set up through which they could address this question. I think having an estimate of the temporal stability of the quantal parameters across boutons would boost the innovation of the conclusions reached by the authors.

Reviewer #2 (Remarks to the Author):

The authors investigate vesicle release and quantal parameters at CA3-CA1 synapses in organotypic hippocampal slice cultures using the glutamate sensor iGluSnFR. They show that: most Schaffer collateral boutons have a single active zone; release is confined to the AZ; under low Pves mainly a single vesicle is released whereas under high Pves multivesicular release occurs; and Pves is variable among boutons.

The work is solid and thorough, technically I have only minor issues/comments (see below). The findings bring relatively little conceptual novelty beyond what is known about these synapses, although the corroboration of previous studies with an independent technique is a valuable contribution. Some parts of the results have been already published by the same group (Dürst et al., 2019 Nature Protocols).

Specific comments:

- Discrimination of successes and failures is based on the rather arbitrary criterion of 2SD. It is not clear whether successes and failures were always clearly distinguishable by this threshold and whether this was independent of the reporter expression level.

- The part raising the most questions is the quantal analysis (fig 7). Several studies pointed out problems arising from fitting amplitude histograms, and some of these problems remain valid here (e.g. limited number of observations, binning the data and the effects of different bin sizes). Distinguishing failures from successes is not convincing in all cases shown (e.g. Fig 7b), and the effect of reporter saturation is only predicted from modelling.

A related question is, how variable is the saturation of the reporter across individual boutons of the same and of different axons? This would be nice to be examined by externally applied glutamate at different concentrations.

- The authors should show correlations among the quantal parameters (not only how they correlate with the mean synaptic strength).

- It was surprising that the quantal parameters in 2 mM Ca²⁺ were not estimated.

- Due to the outliers, it would be better to evaluate correlations (Fig 1, 8) using a nonparametric approach.

- Fig. 4: The numbers on the X axis seem to be shifted in panel b.

Reviewer #3 (Remarks to the Author):

This work is a technical tour de force to cherish. Particularly notable are the careful controls and internal measures of consistency, which vouch the major technical advances. It is clear that the work reports both unquantal and multiquantal release from the same terminals, a real breakthrough. The main figures represent a remarkable series of observations that provide clear answers to long standing questions - where do vesicles fuse, how much glutamate do they release, how does this activate postsynaptic receptors? Unlike most similar work to date on presynaptic release, this is done in slice, not cultured neurons, meaning that the observations are directly applicable to the true physiological situation.

The mathematical modelling is impressive. In a couple of cases, it could be explained or justified a bit better (see below). The group has a reputation for careful quantitative work, and this work is no exception. Leading the average reader more gently into the mathematical and technical details would in turn make the work more accessible, and authoritative.

Major points

1. "Attempts to electrically stimulate individual synapses gave rise to the hypothesis that individual synapses in the central nervous system can only release a single vesicle ('uniquantal release') . With electrophysiology alone, however, it is difficult to distinguish between stimulation of multiple synapses and stimulation of a single synapse that is capable of multivesicular release. I agree with these statements but the connection between them seems to me to be lacking or inverted. If you believe strongly that release is uniquantal, multivesicular release is just multisynapse release. Therefore, any apparent graded nature of quantal release must be attributed to the failure of the electrical stimulation method. There is no dispute here, but I think it could be explained better by an extra sentence mentioning that many signs of MVR are present but it has long been controversial what is going on - clarifying this point is the core of the paper.
2. "In simultaneously imaged neighboring boutons, failures were not correlated, arguing for stochastic glutamate release" - this is a terrifically important observation.
3. "Consequently, iGluSnFR molecules became bound (and fluorescent) and doubly bound AMPARs opened (Fig. 6b)." Reading the text, it is not stated where the idea that receptors are doubly bound comes from. The occupancy of individual synaptic AMPA receptors during synaptic transmission is hard to know but single channel recording tells us that doubly-bound AMPA receptors, even with auxiliary proteins present, will not give much in the way of current. Only triply bound or fully bound receptors open with any appreciable conductance.

In the methods, I was led to the Jonas et al paper. The model in this paper is rather dated and gave rise to some predictions about AMPA receptors that were later shown to be at best unlikely (~100 receptors per quantum etc). It is not clear to me how the use of this model impacts the assertion that, despite their rather different affinities for glutamate, iGluSnFR is reporting AMPA receptor occupancy really well. Numerous great and important questions are tackled in this work but this seems shaky. I would not redo any of these simulations, but I think at least a little comment that other models (like the Hausser and Roth model, which is probably more accurate) might give slightly different answers. Perhaps the authors can provide a bit more perspective here, if they want to back up their assertion. I could well believe that the AMPAR model chosen isn't that important, but that would take some showing (and could detract from the main points of the manuscript).

4. "the putative fusion events were typically localized to a small region " any quantitation possible?
5. "we show that the dynamic range of iGluSnFR is similar to postsynaptic AMPA receptors, although the kinetics of the underlying glutamate transients in the synaptic cleft is an order of magnitude faster "

faster than either? Or [glu] transient is faster than iGluSnFR by only an order of magnitude? Previous calculations would not agree.

6. "Due to vesicle pool depletion and AMPAR desensitization, we expected to see some degree of depression at higher stimulation frequencies, especially under conditions of high release probability (4 mM [Ca²⁺]_i). Considering the entire article, I found this point confusing. Of course, prior work suggests depression, but the convincing mathematical modelling later on shows that P_v is low (at 1 mM) and N is high . Therefore, potentiation is expected (you see it at 1 mM). But, although vesicles should run out with repetitive stimulation at 4 mM Ca, it seems that giving only 2 pulses has little chance to change the

pool. Is N is really docked vesicles? Does Pves change dynamically during stimulation? Perhaps there is a subtlety here that I missed, but I think it would be good to identify what is canonical wisdom and what might be updated by the work in hand.

minar points

It would be good to put a reference to tdimer2. There are quite a few fluorescent proteins with similar names.

In Figure 3b, is the color scale for the gaussian fit the same as for the left panel? Perhaps you don't need two color scales and can change the labelling (move G. fit above)

Figure 7 b and c have some errors - a box over the axis?

Andrew Plested

Reviewer #1 (Remarks to the Author):

A quantitative description of neurotransmitter release was originally developed by Katz using the frog neuromuscular junction as a model system, in the 1950s. Since then, a number of groups have estimated the quantal parameters N , Pr/P_v , and q at peripheral and central synapses. As the authors mention in their introduction, there are a number of limitations to keep in mind when attempting to derive these quantal parameters at small central synapses, which can be largely attributed to electrotonic filtering of synaptic responses arising at a distance from the recording site (typically the soma). In this work, the authors develop a new experimental approach to estimate P_v at Schaffer collateral synapses in organotypic slice cultures. The approach is based on the combined use of optical reporters of extracellular glutamate (iGluSnFR) and **sophisticated high frequency spiral scan two-photon imaging**. The main findings of this work indicate that Schaffer collateral synapses switch from a mostly univesicular to multivesicular release at increasing extracellular calcium concentrations. A similar conclusion was reached by Christie and Jahr (J Neurosci 2006), but this work was solely based on the use of competitive low affinity antagonists for AMPA receptors. For this reason, one could claim that the data from Christie and Jahr provided only an indirect proof of what the current paper shows with much more **direct evidence**. The work is consistent with that of Jensen et al (Nat Comm 2019), in which the authors imaged iGluSnFR and Cal-590 at individual boutons. Overall, the manuscript is clearly presented, the imaging approaches are elegant and rigorous, and the data interpretation is fair. **Where the paper falls short, in my opinion, is novelty**, and I hope the points raised below will ultimately help to overcome this potential limitation.

We do not think our study lacks novelty and would like to point out five aspects that we have not seen published anywhere else:

- 1) We show that reliable extraction of quantal parameters from optical measurements is only possible when a) recordings under radically different conditions are combined, b) photon counting statistics are considered and c) indicator saturation is corrected for. We started our study by recording as many trials as possible from single synapses under constant conditions and tried to extract quantal parameters by histogram fitting (similar to Soares et al. 2019, Front Syn Neurosci). We encountered the problem that many combinations of (p, q, n) exist that provide equally good fits to the data (please see Reviewer Fig. 2, below). The only way to constrain the fit and to narrow it down to a single set of parameters was to optically record the same synapse under very different conditions. This has not been achieved before. In our opinion, it is the best strategy to distinguish e.g. a synapse with 3 release-ready vesicles and $p_{ves}=0.3$ from a synapse with 10 vesicles and $p_{ves}=0.1$ (both of which have $p_{syn}=0.65$). As an added bonus, we could thus demonstrate that every 'univesicular synapse' can be converted to a 'multivesicular synapse' by simply changing the environmental conditions, thus resolving this long-standing controversy.
- 2) We are not aware of any study that has monitored both evoked responses and single vesicle events ('minis') from a single identified Schaffer collateral synapse, as we do in our Sr^{2+} experiments. Seeing the 'quantum' directly provided us with an elegant confirmation of the histogram fitting approach, greatly increasing our confidence in the latter.
- 3) We are first to use Monte Carlo simulations of 3D glutamate diffusion to quantify the orientation-dependence of optical measurements. This source of variability (tilt angle) has been neglected so far.

- 4) We performed two-photon superresolution microscopy of the location of fusing vesicles. This is not a gimmick, as a major criticism of electrophysiological evidence for multivesicular release (e.g. Christie and Jahr 2006) points to the possibility of multiple active zones. For the first time, we show that dramatically changing the release probability does not change the physical position of vesicle release and thus reflects the activity of a single active zone. We have already published the localization method with some examples in Nat Protocols, but the quantitative comparison of the same boutons in high and low Ca is completely new.
- 5) Most importantly, we show that the *vesicular* release probability is highly variable between individual SC boutons, and that this variability is the main determinant of (pre-)synaptic strength. (Not to be confused with the *synaptic* release probability, which has been analyzed in many previous studies, requiring only simple thresholding).

It seems like the quantal amplitude determined using the Sr⁺ approach was only recorded in n=3 boutons. The authors use three methods to measure q, but one is still left to wonder whether the conclusion that bouton-to-bouton variability in q is a side effect of the low sensitivity of the detection approaches. In other words, if an average synaptic vesicle contains 2,000 glutamate molecules, what is the smallest change in the number of these molecules that can be detected by the three approaches described on page 16? Based on these considerations, one cannot exclude the possibility that relying on such a small sample size might introduce errors for the quantal analysis presented in this manuscript. Increasing the sample size would help.

We performed additional Sr²⁺ experiments and increased the sample size to n = 7 (Fig. 5e). The new measurements confirmed the relatively low inter-bouton variability of q. We would like to point out that the residual variability is not a consequence of poor glutamate sensitivity, but is expected due to the different orientation (tilt) of the synaptic cleft in individual experiments (please see our M-Cell simulations, Fig. 5d). The signal-to-noise ratio in these experiments (see single trials in Fig. 3a) was actually very high.

The authors state that a typical bouton (assuming a typical bouton truly exists) displays multivesicular release 4% of times at [Ca²⁺]_e=1 mM, and that this proportion increases to 77% when [Ca²⁺]_e=4 mM. The physiological concentration of calcium in the CSF is thought to be 1.2 mM, and fluctuations in this baseline level can occur as a result of neuronal activity, etc. It would be valuable to take advantage of the model to show how the probability of uni/multivesicular release change over [Ca²⁺]_e ranging from 0-4 mM. How steep is this relationship? Is such a high proportion of multivesicular release (77%) ever reached under physiological conditions, where [Ca²⁺]_e=4 mM is unlikely to be reached?

These are interesting suggestions, but the steepness of the relationship depends on many factors, not just on [Ca²⁺]_e. A simulation in Matlab (Reviewer Fig.1) shows how the fraction of multivesicular events depends on the calcium concentration, but also on the number of release-ready vesicles (N). In vivo, release probability is further subject to neuromodulation, e.g. by adenosine and extracellular K⁺, so we hesitate to claim a specific fraction of multivesicular release as 'physiological' just based on the calcium concentration. We decided to remove these numbers from the manuscript, as indeed a 'typical bouton' does not exist and the *variability* of quantal parameters is the most interesting biological aspect of our study.

Reviewer Figure 1: Matlab simulation of binomial release. At low Ca²⁺, univesicular events dominate while multivesicular events requires higher Ca²⁺ concentrations. The calcium concentration at which 50% of events are multivesicular (dashed line) depends on the number of release-ready vesicles.

The authors conclude that P_{syn} is variable among terminals, and suggest that this could be attributed to differences in the distance between the RRP and calcium channels. This explanation seems a little simplistic, as one would imagine that differences in the number of calcium channels (not just the distance between the RRP and the calcium channels) could also contribute to this effect. Can the authors explain more carefully their reasoning? Are they implying that the RRP and the calcium channels must be separate from each other (e.g. Nadkarni et al, PNAS 2012) or that the calcium channels can be sparse within the domain of the active zone where the synaptic vesicles are docked (e.g. Scimemi and Diamond, 2012)?

We mentioned the calcium channel arrangement as one example how synapses could regulate p_{ves} , but the number of calcium channels is admittedly as important. Our measurements do not help resolving the Levine/Diamond controversy about the location of the calcium channels, but we now make a more balanced statement with a pointer to the very nice simulation study:

“The vesicular release probability was remarkably variable between individual presynaptic terminals, suggesting that the number of calcium channels and/or their distribution with respect to the releasable vesicles is not uniform (Nadkarni 2012).”

The finding that there is variability in P_{syn} from bouton to bouton is not particularly unexpected, but how does the magnitude of this variability change at varying $[Ca^{2+}]_e$? How does $[Ca^{2+}]_e$ affect the variability of P_v within a given bouton?

It is true that variability in P_{syn} was expected, but for the first time, we answer the question: WHY is P_{syn} so variable? This is because P_{ves} is different in different boutons! This is a non-trivial finding (we would have put our money on N , the number of docked and primed vesicles, as the parameter that matters most).

We calculated the coefficient of variation of p_{syn} in low and high Ca²⁺ and appended the text (p.5):

“Since vesicle fusion is Ca²⁺-dependent, we expected a steep dependence of p_{syn} on the extracellular Ca²⁺ concentration, $[Ca^{2+}]_e$. Indeed, switching $[Ca^{2+}]_e$ from 1 mM to 4 mM (Fig.1d) dramatically increased p_{syn} from 0.26 to 0.87 (Fig.1e), with some boutons reaching the ceiling of $p_{syn} = 1$. The variability of p_{syn} was much larger in 1 mM $[Ca^{2+}]_e$ (CV = 0.61) compared to the same synapses in 4 mM $[Ca^{2+}]_e$ (CV = 0.26).”

We cannot assess the potential variability of Pves within a given bouton, so we assume it to be zero (all readily releasable vesicles within one bouton have the same release probability). Every model requires a certain degree of abstraction or else will drown in free parameters.

How stable are the properties of these synapses over time? Clearly, calcium channels have a limited lifetime in the presynaptic active zone, and the authors have an impressive set up through which they could address this question. I think having an estimate of the temporal stability of the quantal parameters across boutons would boost the innovation of the conclusions reached by the authors.

We wish we could perform multiple quantal analyses from individual synapses. Our experiments require continuous whole-cell patch clamp recordings to induce action potentials by somatic current injection with 100% reliability. We find it hard to extend these recordings beyond 45-60 min, the time we need to find distal terminals, collect 200+ trials and change the perfusion solution to record at different [Ca²⁺]_i. As the synaptic release probability is typically fairly stable over the course of the experiment, we suspect that quantal parameters do not change much in a constant environment. We provide in Extended Data Fig. 1 examples of recordings, illustrating the relative stability of p_{ves} over time.

Reviewer #2 (Remarks to the Author):

The authors investigate vesicle release and quantal parameters at CA3-CA1 synapses in organotypic hippocampal slice cultures using the glutamate sensor iGluSnFR. They show that: most Schaffer collateral boutons have a single active zone; release is confined to the AZ; under low Pves mainly a single vesicle is released whereas under high Pves multivesicular release occurs; and Pves is variable among boutons.

The work is **solid and thorough**, technically I have only minor issues/comments (see below). The findings bring **relatively little conceptual novelty** beyond what is known about these synapses, although the corroboration of previous studies with an independent technique is valuable contribution. Some parts of the results have been already published by the same group (Dürst et al., 2019 Nature Protocols).

Yes, we have previously published our innovative spiral scanning approach for fast glutamate imaging in Nature Protocols, but this was a technical protocol that contains no optical quantal analysis, no diffusion modelling and no release desynchronization experiments. We now present a full dataset of 27 analyzed boutons which allowed us to draw biologically relevant conclusions about the variability of quantal parameters in CA1.

To avoid any appearance of self-plagiarism and to shift the focus from technological aspects to the novel biological and theoretical insights, we substantially shortened the description of the spiral scanning approach (merging Figs. 1 and 2) and the fusion site localization procedure (new Fig. 2), avoiding any overlap with Dürst et al., 2019.

Specific comments:

- Discrimination of successes and failures is based on the rather arbitrary criterion of 2SD. It is not clear whether successes and failures were always clearly distinguishable by this threshold and whether this was independent of the reporter expression level.

The two-sigma criterion provides excellent agreement with the more complex failure analysis by fitting a binomial model to all data. It requires discarding experiments that do not reach a

defined SNR criterion (low photon count, see Extended Data Fig. 2). We now provide a comparison between different thresholds in Extended Data Fig. 3 and a detailed explanation to clarify this point.

- The part raising the most questions is the quantal analysis (fig 7). Several studies pointed out problems arising from fitting amplitude histograms, and some of these problems remain valid here (e.g. limited number of observations, binning the data and the effects of different bin sizes). Distinguishing failures from successes is not convincing in all cases shown (e.g. Fig 7b), and the effect of reporter saturation is only predicted from modelling. A related question is, how variable is the saturation of the reporter across individual boutons of the same and of different axons? This would be nice to be examined by externally applied glutamate at different concentrations.

Detailed calibrations of iGluSnFR have been published by Marvin et al. 2013 & 2018. It is not feasible to calibrate iGluSnFR by glutamate application deep in intact tissue, as local glutamate concentrations are strongly affected by EAATs on neurons and glia (pharmacological block of which leads to toxicity and tissue swelling after Glu application). Addressing the question of iGluSnFR saturation during brief (very non-stationary!) glutamate transients in the synaptic cleft was our main motivation to embark on the m-Cell modeling. We believe that modeling the binding process (fixing all parameters based on biophysical measurements) is the best way to derive from our optical measurements the amount of glutamate released into the synaptic cleft. That the indicator itself displays different affinities or kinetics in different neurons or boutons we consider highly unlikely.

As to the effect of bin size on histogram fitting: We actually performed our entire analysis three times, using different bin sizes. We found quantal parameter extraction very robust, with a small (<10%) effect of too wide bins in few synapses. We now report bin size effects in Extended Data Fig. 9. In Reviewer Figure 2 (below), we show the problem of extracting 3 quantal parameters from a single histogram (ambiguity, many good solutions). In contrast, our high/low Ca²⁺ strategy allows fixing q from the low Ca histogram (mostly univesicular responses) and forcing the synapse to reveal its maximum potency in high Ca.

- The authors should show correlations among the quantal parameters (not only how they correlate with the mean synaptic strength).

We now provide a table of correlations between all quantal parameters and short-term plasticity in Extended Data Fig. 10. 'Raw' numerical data from all characterized boutons (large spread sheet) are available upon request. We extended the discussion (last paragraph) to compare to the literature the strong anticorrelation between synaptic strength and PPR we observed in 4 mM Ca²⁺, but not in 1 mM Ca²⁺.

- It was surprising that the quantal parameters in 2 mM Ca²⁺ were not estimated.

One of our key insights during this study was that full quantal analysis is only possible when boutons are recorded two very different conditions. The reason is that in 2 mM Ca²⁺, a (simulated) bouton with a low number of high p_{ves} vesicles produces histograms very similar to a bouton with a high number of low p_{ves} vesicles (Reviewer Fig. 2). Thus, only a simple failure analysis is possible in 2 mM Ca²⁺ (which we present in Fig. 1g).

Reviewer Figure 2: Quantal analysis of data recorded in one condition only (2mM Ca²⁺) produces ambiguous results. The error function (lower left) is equally low (dark blue) for many combinations of n and p_{ves} . Lower right panel shows probability density function for a good fit with the least number of vesicles ($n = 3$), but other combinations of q , n and p_{ves} produce very similar looking predictions. The ambiguity is resolved when two histograms (high and low Ca²⁺) are available from the same bouton (see Fig. 6).

- Due to the outliers, it would be better to evaluate correlations (Fig 1, 8) using a nonparametric approach.

We now performed non-parametric (Spearman) correlations tests for all parameter pairs (extracted quantal parameters and PPRs), results are presented in the new Extended Data Figure 10. The results confirm our previous analysis using linear regression, but reach higher levels of significance. Thank you for this helpful suggestion! Based on the new analysis, we now discuss the anti-correlation between synaptic strength and PPF in high Ca²⁺ which adds an interesting aspect to our study (p. 20):

“The use of available resources (the readily releasable pool of vesicles) is expected to affect the short-term plasticity of a synapse (Dobrunz & Stevens; Debanne et al.). We found that Schaffer collateral synapses that released little glutamate in response to a single AP produced pronounced paired-pulse facilitation in 4 mM [Ca²⁺]_e (Extended Data Fig. 10), consistent with a previous study on cultured hippocampal neurons (Sakamoto et al.). A recent study (Jensen et al.) combined glutamate and calcium imaging at individual boutons in 2 mM [Ca²⁺]_e and found short-term plasticity to be independent of bouton [Ca²⁺]_i. We found that in 1 mM [Ca²⁺]_e, correlations between PPR and quantal parameters disappeared (Extended Data Fig. 10), although PPR at individual synapses was quite variable under these conditions (Fig. 5f). We conclude that short term plasticity and differential filtering of high frequency signals as described by Markram and Tsodyks can be readily linked to quantal parameters in high Ca²⁺, but may be dominated by other factors under more physiological, low-use conditions. Proximity to astrocytic processes may play a role (Sibille et al.) as well as the age of the synapse, as mature synapses are able to use their vesicle complement much more efficiently (Rose et al.).”

- Fig. 4: The numbers on the X axis seem to be shifted in panel b.

Thank you! They are now in the correct positions.

Reviewer #3 (Remarks to the Author):

This work is a **technical tour de force** to cherish. Particularly notable are the careful controls and internal measures of consistency, which vouch the **major technical advances**. It is clear that the work reports both unquantal and multiquantal release from the same terminals, a **real breakthrough**. The main figures represent a remarkable series of observations that provide **clear answers to long standing questions** - where do vesicles fuse, how much glutamate do they release, how does this activate postsynaptic receptors? Unlike most similar work to date on presynaptic release, this is done in slice, not cultured neurons, meaning that the observations are directly applicable to the true physiological situation.

The mathematical modelling is **impressive**. In a couple of cases, it could be explained or justified a bit better (see below). The group has a reputation for **careful quantitative work**, and this work is no exception. Leading the average reader more gently into the mathematical and technical details would in turn make the work more accessible, and authoritative.

Major points

1. "Attempts to electrically stimulate individual synapses gave rise to the hypothesis that individual synapses in the central nervous system can only release a single vesicle ('uniquantal release'). With electrophysiology alone, however, it is difficult to distinguish between stimulation of multiple synapses and stimulation of a single synapse that is capable of multivesicular release."

I agree with these statements but the connection between them seems to me to be lacking or inverted. If you believe strongly that release is uniquantal, multivesicular release is just multisynapse release. Therefore, any apparent graded nature of quantal release must be attributed to the failure of the electrical stimulation method. There is no dispute here, but I think it could be explained better by an extra sentence mentioning that many signs of MVR are present but it has long been controversial what is going on - clarifying this point is the core of the paper.

Thank you for this suggestion. We expanded the section:

"Attempts to electrically stimulate individual synapses gave rise to the hypothesis that individual synapses in the central nervous system can only release a single vesicle ('uniquantal release') (Redman, 1990). Graded postsynaptic responses were not considered as hallmark of multivesicular release events, but attributed to a failure of single-synapse stimulation (Dobrunz & Stevens, 1997). As electrophysiology does not provide spatial information about the origin of the signals, it was not possible to distinguish between the two rival interpretations at the time."

2. "In simultaneously imaged neighboring boutons, failures were not correlated, arguing for stochastic glutamate release" - this is a terrifically important observation.

Thank you!

3. “Consequently, iGluSnFR molecules became bound (and fluorescent) and doubly bound AMPARs opened (Fig. 6b).”

Reading the text, it is not stated where the idea that receptors are doubly bound comes from. The occupancy of individual synaptic AMPA receptors during synaptic transmission is hard to know but single channel recording tells us that doubly-bound AMPA receptors, even with auxiliary proteins present, will not give much in the way of current. Only triply bound or fully bound receptors open with any appreciable conductance.

We apologize for the poorly phrased summary of the glutamate AMPAR reaction scheme that was used for modeling. As the panel displays the formation of open state AMPARs over time, we rephrased to

“Consequently, iGluSnFR molecules became bound (and fluorescent) and AMPARs opened (Fig. 4b)”

In the methods, I was led to the Jonas et al paper. The model in this paper is rather dated and gave rise to some predictions about AMPA receptors that were later shown to be at best unlikely (~100 receptors per quantum etc). It is not clear to me how the use of this model impacts the assertion that, despite their rather different affinities for glutamate, iGluSnFR is reporting AMPA receptor occupancy really well. Numerous great and important questions are tackled in this work but this seems shaky. I would not redo any of these simulations, but I think at least a little comment that other models (like the Häusser and Roth model, which is probably more accurate) might give slightly different answers. Perhaps the authors can provide a bit more perspective here, if they want to back up their assertion. I could well believe that the AMPAR model chosen isn't that important, but that would take some showing (and could detract from the main points of the manuscript).

As in our simulation, all glutamate molecules are released at once, the additional two desensitized conformations that the Häusser model adds to the Jonas model are not expected to have a dramatic effect on our simulations. We consider the correction for non-linearity of the indicator as very important for accurate quantal analysis, and previous studies using GluSnFR have neglected this. With respect to the absolute level of AMPAR saturation, the point is well taken and we added the following ‘disclaimer’ (p.10):

“Admittedly, real AMPARs display desensitization, multiple conductance levels and complex gating modes (Prieto & Wollmuth 2010) that are not captured in our highly simplified model. For quantal analysis, however, it is conceptually important to consider that iGluSnFR is not a linear reporter of cleft glutamate, but has a strongly saturating response similar to postsynaptic AMPARs.”

4. “the putative fusion events were typically localized to a small region” any quantitation possible?

The numbers are now included in the legend of Fig. 2:

“d) To quantify the spatial distribution of release events at each bouton, an ellipse was fit to contain 95% of success locations. Both long and short axis of the ellipse were significantly smaller (Wilcoxon test, $p = 0.04$, $n = 12$ boutons) under high Ca^{2+} conditions ($0.21 \pm 0.03 \mu\text{m}$; $0.41 \pm 0.06 \mu\text{m}$) compared to $1 \text{ mM } [\text{Ca}^{2+}]_e$ ($0.28 \pm 0.01 \mu\text{m}$; $0.56 \pm 0.06 \mu\text{m}$).”

5. “we show that the dynamic range of iGluSnFR is similar to postsynaptic AMPA receptors, although the kinetics of the underlying glutamate transients in the synaptic cleft is an order of magnitude faster”; faster than either? Or [glu] transient is faster than iGluSnFR by only an order of magnitude? Previous calculations would not agree.

Thank you for pointing this out this mistake! As cleft [Glu] is down to 50% 23 μ s after release, we rephrased to:

“Using dual patch-clamp recordings and Monte Carlo simulations of glutamate diffusion, we show that the dynamic range of iGluSnFR is similar to postsynaptic AMPA receptors, although the kinetics of the underlying glutamate transient is 2-3 orders of magnitude faster than the iGluSnFR response. Thus, iGluSnFR signals are a good proxy for postsynaptic responses, but do not report the speed of glutamate diffusion out of the synaptic cleft.”

6. “Due to vesicle pool depletion and AMPAR desensitization, we expected to see some degree of depression at higher stimulation frequencies, especially under conditions of high release probability (4 mM [Ca²⁺]_e).”

Considering the entire article, I found this point confusing. Of course, prior work suggests depression, but the convincing mathematical modelling later on shows that P_{ves} is low (at 1 mM) and N is high. Therefore, potentiation is expected (you see it at 1 mM). But, although vesicles should run out with repetitive stimulation at 4 mM Ca, it seems that giving only 2 pulses has little chance to change the pool. Is N is really docked vesicles? Does P_{ves} change dynamically during stimulation? Perhaps there is a subtlety here that I missed, but I think it would be good to identify what is canonical wisdom and what might be updated by the work in hand.

We chose a more neutral lead-in to this paragraph and tried to better explain the results, which are consistent with previous interpretations from electrophysiology and competitive antagonist experiments (Debanne et al.) and our earlier high-frequency experiments (Helassa 2018). Again, we would argue that optical glutamate measurements monitor presynaptic plasticity much more directly than electrophysiological recordings, but we don't want to spell this out this in every section of the manuscript.

“We were interested how the dramatically boosted synaptic transmission in 4 mM [Ca²⁺]_e would change short-term plasticity at this synapse. We compared paired-pulse ratios ...

It is remarkable that Schaffer collateral synapses are able to maintain a fairly linear paired pulse response over a ~10-fold modulation in synaptic strength. As previously suggested, the depletion of the readily-releasable pool of vesicles in high Ca²⁺ conditions seems to be largely compensated by Ca²⁺-dependent facilitation of release of the remaining docked vesicles (Debanne et al.). During prolonged high frequency trains, however, this compensatory mechanism runs out of resources, resulting in strong depression (Helassa et al.).”

minor points

It would be good to put a reference to tdimer2. There are quite a few fluorescent proteins with similar names.

Done. (Campbell et al. 2002)

In Figure 3b, is the color scale for the gaussian fit the same as for the left panel? Perhaps you don't need two color scales and can change the labelling (move G. fit above)

We removed the color-coded Gaussian fits as they appeared in our previous publication in Nature Protocols (an overlap that was criticized by Reviewer 1 and the reviewing editor).

Figure 7 b and c have some errors - a box over the axis?

The probability density functions (integral equal to 1) were amplitude-scaled to match the histograms. We now labeled the y-axis 'Probability' and put a box around the extracted parameters (low / high Ca) to prevent confusion with axis labels.

Reviewers' comments:

Reviewer #1 (Remarks to the Author):

The authors have provided thorough and comprehensive answers to the concerns raised in my first evaluation of the manuscript. In particular, I appreciate the clarification about novelty aspects of this work, which remained unclear in the first submission.

Reviewer #2 (Remarks to the Author):

The authors addressed some of the raised questions. However, not all of my concerns have been sufficiently resolved.

1) Authors' response: "Yes, we have previously published our innovative spiral scanning approach for fast glutamate imaging in Nature Protocols, but this was a technical protocol that contains no optical quantal analysis, no diffusion modelling and no release desynchronization experiments. We now present a full dataset of 27 analyzed boutons which allowed us to draw biologically relevant conclusions about the variability of quantal parameters in CA1.

To avoid any appearance of self-plagiarism and to shift the focus from technological aspects to the novel biological and theoretical insights, we substantially shortened the description of the spiral scanning approach (merging Figs. 1 and 2) and the fusion site localization procedure (new Fig. 2), avoiding any overlap with Dürst et al., 2019."

Overlaps with Dürst et al 2019 in the previous version of the manuscript included several identical figure panels. While some of the duplications have been removed, others remained, such as in Fig. 1b (Dürst 2019 Fig 5f) and in Extended Data Fig. 4. (Dürst 2019 Fig. 6).

Extended Data Fig 4 is more problematic, since it is relevant to the biological results. In addition, it seems that although panels a-b are identical to that published, and seemingly the same analysis has been done on the same experiments as in the previous paper, the analyzed data and the statistical conclusions have apparently changed.

2) Comment: The part raising the most questions is the quantal analysis (fig 7). Several studies pointed out problems arising from fitting amplitude histograms, and some of these problems remain valid here (e.g. limited number of observations, binning the data and the effects of different bin sizes). Distinguishing failures from successes is not convincing in all cases shown (e.g. Fig 7b), and the effect of reporter saturation is only predicted from modelling. A related question is, how variable is the saturation of the reporter across individual boutons of the same and of different axons? This would be nice to be examined by externally applied glutamate at different concentrations.

Authors' response: "Detailed calibrations of iGluSnFR have been published by Marvin et al. 2013 & 2018. It is not feasible to calibrate iGluSnFR by glutamate application deep in intact tissue, as local glutamate concentrations are strongly affected by EAATs on neurons and glia (pharmacological block of which leads to toxicity and tissue swelling after Glu application). Addressing the question of iGluSnFR saturation during brief (very non-stationary!) glutamate transients in the synaptic cleft was our main motivation to embark on the m-Cell modeling. We believe that modeling the binding process (fixing all parameters based on biophysical measurements) is the best way to derive from our optical measurements the amount of glutamate released into the synaptic cleft. That the indicator itself displays different affinities or kinetics in different neurons or boutons we consider highly unlikely."

Marvin et al 2013 indicates that saturated DF/F0 values can vary in different preparations or in different ROIs. In Dürst et al 2019 Fig. 7, summated release saturates iGluSnFR at around 200% DF/F0 in the same bouton type tested here.

Since the binomial analysis is affected by the saturation value, I maintain that it would be best to try to

measure experimentally the actual value and variability of this parameter under the current conditions. At the least, the authors should calculate how sensitive the results (like N) is on the saturation value of iGluSnFR. It makes a big difference regarding the final conclusions of the manuscript whether changing the saturation value from 440 to e.g. 200 % alters the estimated N by only 1 (e.g. from 5 to 6) or by 3-fold (e.g from 5 to 15).

Minor:

6) Figure 1g, revised legend: "The amplitude of success trials was similar at boutons with $p_{syn} > 0.5$." Is it supposed to be $p_{syn} < 0.5$?

Reviewer #3 (Remarks to the Author):

I am grateful for the authors' careful and thoughtful response. Additional experiments and analysis are accompanied by delicate and fair additions to the text.

I have carefully reviewed manuscript and the answers to the critical points made by all the reviewers. I found the responses and evidence given by the authors highly convincing. I feel that the authors have done the very best job they can to answer difficult questions with this manuscript. In its revised form, this manuscript provides an absolutely authoritative description of vesicle release at this classic synapse.

It was a pleasure to review this manuscript and I look forward to seeing it in print soon.

Andrew Plested

REVIEWER COMMENTS

Author's responses are in bold.

Reviewer #1 (Remarks to the Author):

The authors have provided thorough and comprehensive answers to the concerns raised in my first evaluation of the manuscript. In particular, I appreciate the clarification about novelty aspects of this work, which remained unclear in the first submission.

Reviewer #2 (Remarks to the Author):

The authors addressed some of the raised questions. However, not all of my concerns have been sufficiently resolved.

1) Authors' response: "Yes, we have previously published our innovative spiral scanning approach for fast glutamate imaging in Nature Protocols, but this was a technical protocol that contains no optical quantal analysis, no diffusion modelling and no release desynchronization experiments. We now present a full dataset of 27 analyzed boutons which allowed us to draw biologically relevant conclusions about the variability of quantal parameters in CA1.

To avoid any appearance of self-plagiarism and to shift the focus from technological aspects to the novel biological and theoretical insights, we substantially shortened the description of the spiral scanning approach (merging Figs. 1 and 2) and the fusion site localization procedure (new Fig. 2), avoiding any overlap with Dürst et al., 2019."

Overlaps with Dürst et al 2019 in the previous version of the manuscript included several identical figure panels. While some of the duplications have been removed, others remained, such as in Fig. 1b (Dürst 2019 Fig 5f) and in Extended Data Fig. 4. (Dürst 2019 Fig. 6).

We now show a different, unpublished example experiment in Fig 1b and we have completely removed Extended Data Fig. 4.

Extended Data Fig 4 is more problematic, since it is relevant to the biological results. In addition, it seems that although panels a-b are identical to that published, and seemingly the same analysis has been done on the same experiments as in the previous paper, the analyzed data and the statistical conclusions have apparently changed.

The reviewer is absolutely correct that the statistical analysis in the Extended Data Fig. 4 (now removed) had a different outcome from the one we published in the 2018 Protocols paper. The difference is that we used a 4-sigma threshold to distinguish failures from responses in the Nat. Protocols paper and a 2-sigma threshold in the new analysis. This leads to a slightly different shape of the Gaussian fit to the scrambled data, which in this case (p-values very close to the 0.05 threshold) made a difference which parameter is significantly different and which one is not. (For transparency, we include the raw data, extracted Pr values, and Matlab function to randomize the bouton pairs). Our wording in the Protocols paper reflects the borderline significance of both amplitude and release probability differences at neighboring boutons, but we did not state the threshold correctly (We have since notified Nat. Protocols about this error). As the analysis of neighboring boutons was already presented in our Protocols paper, we removed Extended Data Fig. 4 to avoid any overlap and now just cite our publication.

A failure analysis approach is always sensitive to the arbitrary threshold. This motivated us to develop the histogram approach, combining information from high and low Ca²⁺ conditions. In Extended Data Fig. 3, we now show that a failure analysis with 2-sigma threshold is in good agreement with the release probabilities from our histogram analysis (but slightly worse for a 4-sigma threshold). The histogram analysis and its results are central and novel parts of this study. For experts, it is of interest how the parameters extracted by the new method compare to the much simpler failure analysis (thus Extended Data Fig. 3).

2) Comment: The part raising the most questions is the quantal analysis (fig 7). Several studies pointed out problems arising from fitting amplitude histograms, and some of these problems remain valid here (e.g. limited number of observations, binning the data and the effects of different bin sizes).

We are not aware of any study that, as ours, combines the results of high and low Ca^{2+} measurements on single synapses for a quantal analysis by histogram fitting. Yes, attempts to fit a single histogram, in one condition using three (p , q , n) or even more free parameters (multi-Gaussian fits without binomial restrictions) have indeed not been very convincing because, as the reviewer correctly points out, many parameter combinations produce near-identical histograms (underdetermined system). This is precisely where our approach is better, first using the low Ca^{2+} condition and histogram to determine q , leaving only two free parameters to fit the high Ca^{2+} histogram. Importantly, the width of the Gaussian peaks is not treated as a free parameter, but was calculated from the photon shot noise in the baseline. Due to these improvements, we are not dealing with an underdetermined system, but can extract quantal parameters with high confidence. Pointing to critical comments about previous studies does not do justice to our methodical improvements.

Given the limited number of observations possible, our high/low Ca^{2+} approach was considered optimal by Gary Bhumbra, a quantal analysis expert at UCL we consulted. With regard to the specific criticism about binning, we show that our quantal analysis is robust to bin size (Extended Data Figure 8).

Distinguishing failures from successes is not convincing in all cases shown (e.g. Fig 7b), and the effect of reporter saturation is only predicted from modelling.

Distinguishing failures from successes is indeed a major problem for traditional failure analysis with noisy signals, which is why we consider threshold-based failure analysis to be superseded by our quantal parameter fitting procedure. Even in cases where there is no clear separation of failures and responses in 1 mM Ca^{2+} , we can safely assume that under these low Pr conditions mostly single vesicles are released. Furthermore, the failures are normally distributed (symmetrically around zero). Therefore, fitting two Gaussians gives a reasonable estimate of the quantal amplitude even if some individual responses cannot be reliably classified as failures or successes.

The reporter saturation level has been measured *in vitro* by application of 10 mM of glutamate to iGluSnFR protein (Helassa N et al., 2019; Marvin JS et al., 2013). The largest single trial response we measured in life synapses (Reviewer Fig. 1) was 434% df/f (in 4 mM Ca^{2+}), approaching the *in vitro* value of 440%. Thus, *in vivo* and *in vitro* measurements are in good agreement.

Reviewer Figure 1: Distribution of the largest single trial response in 4 mM $[\text{Ca}^{2+}]_e$ for every single bouton in our sample ($n = 27$ boutons).

A related question is, how variable is the saturation of the reporter across individual boutons of the same and of different axons? This would be nice to be examined by externally applied glutamate at different concentrations.

We tried to perform the suggested calibration experiments, but strong swelling of the boutons made quantitative measurements impossible (Reviewer Fig. 2). The optical properties of the tissue above the imaged boutons also changed, altering laser excitation of fluorescence in unpredictable ways.

Reviewer Figure 2: Bath application of 10 mM glutamic acid leads to dramatic swelling of the bouton, changing the concentration of cytoplasmic fluorescent molecules (tdimer-2, upper row) and iGluSnFR on the surface (lower row). NBQX, CPP and TTX were added to block synaptic transmission and action potentials. Image stacks were taken every 2 min.

Authors' response: "Detailed calibrations of iGluSnFR have been published by Marvin et al. 2013 & 2018. It is not feasible to calibrate iGluSnFR by glutamate application deep in intact tissue, as local glutamate concentrations are strongly affected by EAATs on neurons and glia (pharmacological block of which leads to toxicity and tissue swelling after Glu application). Addressing the question of iGluSnFR saturation during brief (very non-stationary!) glutamate transients in the synaptic cleft was our main motivation to embark on the m-Cell modeling. We believe that modeling the binding process (fixing all parameters based on biophysical measurements) is the best way to derive from our optical measurements the amount of glutamate released into the synaptic cleft. That the indicator itself displays different affinities or kinetics in different neurons or boutons we consider highly unlikely."

Marvin et al 2013 indicates that saturated DF/F0 values can vary in different preparations or in different ROIs. In Dürst et al 2019 Fig. 7, summated release saturates iGluSnFR at around 200% DF/F0 in the same bouton type tested here.

This experiment (Dürst et al 2019) was done in 4 Ca²⁺, where there is strong depression of glutamate release at 100 Hz. From the reduced amplitude of the 2nd and subsequent events, it cannot be concluded that iGluSnFR saturation has been reached at this synapse. The "physiological maximum response" of individual boutons (Reviewer Fig. 1) depends on the maximal number of vesicles that were released in a single trial. This number should not be taken as a measure of iGluSnFR saturation at that synapse.

Since the binomial analysis is affected by the saturation value, I maintain that it would be best to try to measure experimentally the actual value and variability of this parameter under the current conditions. At the least, the authors should calculate how sensitive the results (like N) is on the saturation value of iGluSnFR. It makes a big difference regarding the final conclusions of the manuscript whether changing the saturation value from 440 to e.g. 200 % alters the estimated N by only 1 (e.g. from 5 to 6) or by 3-fold (e.g. from 5 to 15).

The properties of individual iGluSnFR molecules (resting fluorescence, Glu-bound fluorescence) will not be different in different boutons. However, this critical comment made us realize that we should use our diffusion model to explore the saturation process in more detail. When imaging an iGluSnFR-expressing bouton during vesicular release, every iGluSnFR molecule experiences a different glutamate transient, dependent on its distance from the fusion site. Distant (extrasynaptic) iGluSnFR molecules “see” much lower Glu concentrations, leading to a lower apparent affinity of the bouton as a whole. (corresponding to apparent saturation values above 440%, green hyperbola in Reviewer Fig. 3). The tilt of the synaptic cleft had relatively minor effects on the saturation curve (Fig. 5f). We conclude that our previous saturation correction based on bath-applied glutamate to iGluSnFR-expressing HEK cells (440% saturation, red hyperbola in Reviewer Fig. 3), was too strong.

Reviewer Figure 3: Hyperbolic curve with asymptotic value 440% (red) is not a good fit for simulated release of 1 – 15 vesicles (black markers, from Monte Carlo simulations). Best hyperbolic fit (green curve) saturates at 600%. We call this value ‘apparent saturation’, as $dF/F = 600\%$ cannot be reached.

Based on these insights, we decided to extract a new set of quantal parameters using the apparent saturation value of a hyperbolic fit to the diffusion simulations (green curve in Reviewer Figure 3, Fig. 5f, Fig. 6).

So, how sensitive is our quantal analysis to the exact value of the saturation correction?

We repeated the entire parameter extraction procedure for apparent saturation values of 600% and 700%, corresponding to an optimally oriented and a 40° tilted synaptic cleft, respectively (Fig. 5f, Fig. 6). A comparison to our previous saturation value of 440% is presented in the new Extended Data Figure 9. Quantal parameter values in individual synapses slightly changed with the assumed saturation value. For example, the median number of releasable vesicles (N) changed from 4 to 3. We extended the discussion (page 19) to better explain the intricacies of the saturation correction to the readers.

How does the new analysis affect our central claim, that the vesicular release probability sets synaptic strength in low Ca^{2+} while N becomes more important under high p_{syn} conditions?

We updated Fig. 7 with the data from the new analysis. Our efforts to get the saturation correction exactly right paid off: According to the new analysis, p_{ves} explains 85% of the variability in strength (up from 66% in the previous analysis). We also updated Extended Data Figure 10 to show Spearman’s r for all parameter combinations. We thank the reviewer for challenging our assumptions, leading to a more extensive exploration of the parameter space. For complete transparency, we will deposit all raw data on Zenodo and all software on GitHub.

Minor:

6) Figure 1g, revised legend: "The amplitude of success trials was similar at boutons with $psyn > 0.5$." Is it supposed to be $psyn < 0.5$?

We corrected the typo, thank you!

Reviewer #3 (Remarks to the Author):

I am grateful for the authors' careful and thoughtful response. Additional experiments and analysis are accompanied by delicate and fair additions to the text.

I have carefully reviewed manuscript and the answers to the critical points made by all the reviewers. I found the responses and evidence given by the authors highly convincing. I feel that the authors have done the very best job they can to answer difficult questions with this manuscript. In its revised form, this manuscript provides an absolutely authoritative description of vesicle release at this classic synapse.

It was a pleasure to review this manuscript and I look forward to seeing it in print soon.

Andrew Plested

Reviewers' comments:

Reviewer #2 (Remarks to the Author):

The authors carefully addressed the raised questions; the corrections and the extended analysis on reporter saturation are satisfactory. I have no further major concerns.

Two minor notes:

- I assume that Extended Data Figure 8 should also be updated as part of the new analysis.
- In Fig 1c middle, the scale bar of 20 ms is missing.